# Heterogeneous preferences and asymmetric insights for AI use among welfare claimants and non-claimants

**Mengchen Dong** [1] ✉, **Jean-François Bonnefon** [2] **& Iyad Rahwan** [1]

The deployment of AI in welfare benefit allocation accelerates decision-making but has led to unfair denials and false fraud accusations. In the US and UK ($N = 3249$), we examine public acceptability of speed-accuracy trade-offs among claimants and non-claimants. While the public generally tolerates modest accuracy losses for faster decisions, claimants are less willing to accept AI in welfare systems, raising concerns that using aggregate data for calibration could misalign policies with the preferences of those most affected. Our study further uncovers asymmetric insights between claimants and non-claimants. Non-claimants overestimate claimants' willingness to accept speed-accuracy trade-offs, even when financially incentivized for accurate perspective-taking. This suggests that policy decisions aimed at supporting vulnerable groups may need to incorporate minority voices beyond popular opinion, as non-claimants may not easily understand claimants' perspectives.

The use of Artificial Intelligence (AI) is becoming commonplace in government operations[1–4]. In the United States alone, a 2020 survey of 142 federal agencies found that 45% were using or planning to use machine learning algorithms to streamline their operations, increase their capacities, or improve the delivery of their public services[2]. In the specific context of providing welfare benefits, the main promise of AI is to speed up decisions[1,5,6]. For many individuals and families, welfare benefits provide critical assistance in times of financial hardship or emergency. Using AI to speed up decisions can avoid delays that would exacerbate these hardships, and decrease the period of uncertainty and anxiety during which applicants are waiting for a decision. However, there is a documented risk that since welfare AI systems often focus on fraud detection, their speed gains come with a biased accuracy loss, increasing the rate at which people are unfairly denied the welfare benefits they are entitled to[5–10].

Given the practical relevance of speed and accuracy for welfare decisions, these performance metrics are often highlighted in public-facing government reports after deploying AI systems, based on the implicit assumption that the general public values speed and accuracy in government services. For example, before the Royal Commission

into the notorious Robodebt scheme in Australia, the government annual report 2019–2020 stated that "the agency automated the assessment and processing of most claims for services", "we processed 1.3 million JobSeeker claims in 55 days, a claim volume normally processed in two and a half years", and "the agency recorded 276,589 feedback contacts...dissatisfaction with a decision, outcome or payment, including waiting too long, not receiving a payment, and rejection of an application or claim (32.1 percent)"[11]. Similarly, the UK Department for Work and Pensions Annual Report stated that with "[i]ncreased use of data analytics and greater automation", they had "146,000 claims checked by Enhanced Checking Service", among which "87,000 check result in change to award"[12]. Although these reports do not explicitly state whether and how welfare AI systems trade off between speed and accuracy, government agencies that seek to deploy welfare AI systems in an acceptable and trustworthy way may benefit from carefully considering public preferences when balancing speed gains and accuracy losses.

Developing AI systems for social good requires not only technological progress but also the integration of a broader set of ethical, legal, and societal considerations, which necessitates incorporating

[1]Center for Humans and Machines, Max Planck Institute for Human Development, Berlin, Germany. [2]Department of Social and Behavioral Sciences, Toulouse School of Economics, Centre National de la Recherche Scientifique (Toulouse School of Management Research), University of Toulouse Capitole, Toulouse, France. ✉e-mail: dong@mpib-berlin.mpg.de

the perspectives of various direct and indirect stakeholders[13]. Caseworkers, developers, and program managers can develop an understanding of the needs and pain points of users of government AI systems through exposure to diverse user cases and civic discussion[14,15]. However, their technical-rational perspective may lead them to overemphasize certain performance metrics while overlooking the perspectives of the general public[14,16]. Public deliberation also plays a crucial role in ensuring that AI systems align with societal values and are perceived as fair and legitimate. While public preferences may not directly dictate AI design choices, they influence the legal and regulatory environment in which AI systems operate, shaping AI development and deployment through political decision-making processes. A prominent example here is the public engagement in the formulation and regulation of autonomous vehicles (AVs). Concerns about disproportionate harm to vulnerable road users and ethical decision-making in crash scenarios have gained significant public and media attention, prompting policies focused on transparency, explainability, and accountability of AV behavior[17].

In the context of social welfare distribution, public preferences should be valued for at least two reasons. First, we know that people who lose trust in the AI used by one government agency also lose trust in the AI used by other government agencies – if welfare AI systems ignore public preferences when balancing speed and accuracy, they risk creating distrust that can bleed into perceptions of other government services[18,19]. Second, and more immediately, the wrong balance of speed gains and accuracy losses could erode the trust of people who need welfare benefits, and make them less likely to apply, for fear of being wrongly accused of fraudulent claims[18], especially when the AI system is labeled with foreboding names like 'FraudCaster'[20] or described as a 'suspicion machine' in the media[8]. In sum, it is important for welfare AI systems to trade off speed and accuracy in a way that is aligned with the preferences of the general public as well as with the preferences of potential claimants.

Great efforts have been made to understand people's attitudes toward and concerns about welfare AI systems, often focusing on the opinions of the general public[18,21] or vulnerable populations directly affected by welfare AI systems[6,22]. Qualitative evidence has also been accumulated regarding the divergent preferences of different stakeholders involved in AI governing systems[3,23], contributing to long-lasting philosophical and regulatory discussions on fairness and alignment principles[24–27]. However, less is known about the extent of divergence in AI performance preferences and reconciliation between different perspectives and interests.

Here we show experimental evidence on two critical challenges for aligning AI with human values in welfare AI systems. First, we identify heterogeneous preferences of welfare claimants versus non-claimants, with claimants showing a stronger AI aversion irrespective of how AI trades off speed and accuracy. Second, we find that welfare claimants' estimates of non-claimants' preferences are closer to the truth than the reverse, suggesting an asymmetry in perspective-taking accuracy. In other words, the perspective of non-claimants is relatively easy to understand, but only claimants understand their own perspective. These results hold in three studies, with a representative US sample and targeted samples balancing the number of claimants and non-claimants in the US and UK. The combination of heterogeneous preferences and asymmetric insights creates the risk of welfare AI systems being aligned with the position of the largest, best understood, least vulnerable group – silencing the voice of the smallest, least understood, most vulnerable group, which nevertheless comprises the primary stakeholders in the deployment of welfare AI.

## Results

### The US representative-sample study

Participants in this study ($N = 987$, representative on age, sex, and ethnicity, 20% self-declaring as welfare claimants) indicated their preference between human and AI welfare decisions. We varied the information about speed gains (1/2/3/4/5/6 weeks faster, as compared to a baseline waiting time of 8 weeks if handled by public servants) and accuracy losses (5/10/15/20/25/30% more false rejections than public servants) within a realistic range, based on governmental reports and third-party investigations[9,28–30], yielding 36 trade-offs (as illustrated in Fig. 1). In each trade-off condition, participants indicated their preference on a scale ranging from 0 = definitely a public servant to 100 = definitely the AI program. Participants were randomly assigned to respond from their own perspective as claimants or non-claimants, or to adopt the opposite perspective.

When participants responded from their own perspective ($N = 506$), their willingness to let AI make decisions was influenced

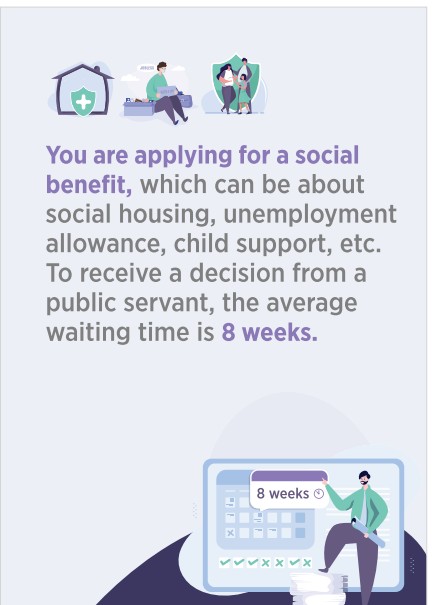

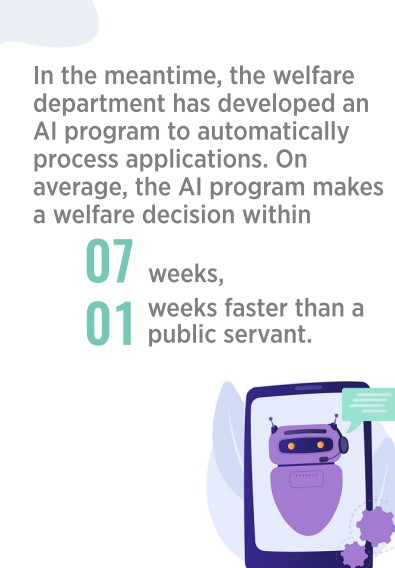

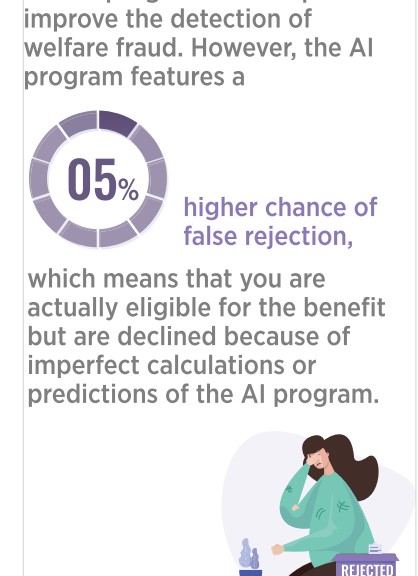

**Fig. 1 | An example of experimental stimuli.** In this example, the AI system is one week faster than humans but leads to a 5% accuracy loss. The complete list of stimuli consisted of 36 such trade-offs, combining speed gains of 1 to 6 weeks (by the increment of 1) and accuracy losses from 5% to 30% (by the increment of 5%).

## Preferences for speed accuracy tradeoffs in the representative US sample

**(A)** Average preferences in the tradeoff conditions

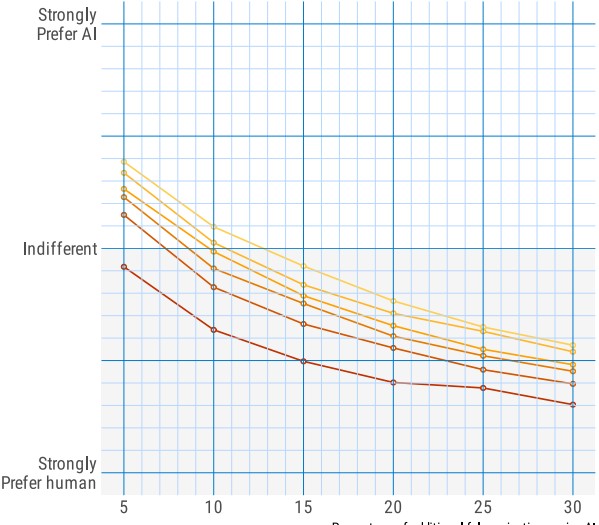

**(B)** Preference distributions in the tradeoff conditions

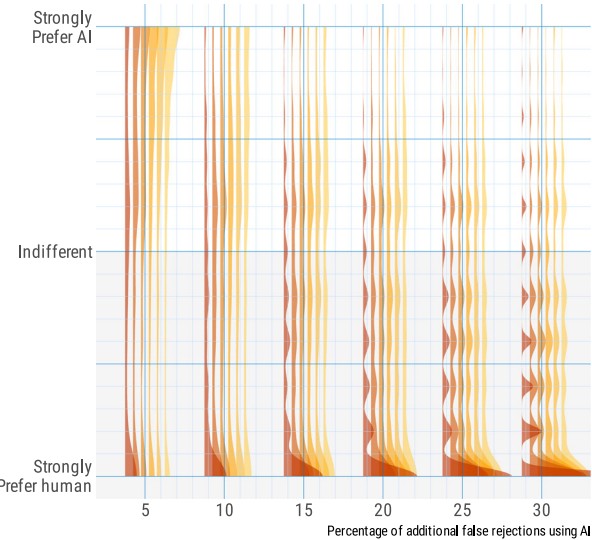

AI speed gain — 6 weeks faster — 4 weeks faster — 2 weeks faster — 5 weeks faster — 3 weeks faster — 1 week faster

**Fig. 2 | Preferences for speed-accuracy trade-offs from own perspective, in the US representative-sample study (N = 506; 21% as welfare claimants). A** On average, respondents trade speed and accuracy at the value of 2.4 percent additional false rejections for speeding up the decision by 1 week. **B** Each slab shows the full distribution of individual preferences in each condition, grouped by speed gain.

both by speed gains, $\beta = 0.19$, $t(17{,}710) = 44.41$, $p < 0.001$, 95% CI [0.18, 0.20], and accuracy losses, $\beta = 0.40$, $t(17{,}710) = 92.35$, $p < 0.001$, 95% CI [0.39, 0.40]. Overall (see Fig. 2A), they traded off a 1-week speed gain for a 2.4 percentage point loss of accuracy. Among these US participants, 21% self-declared as welfare claimants. For all the 36 trade-offs, these claimants (vs. non-claimants) showed greater average aversion to letting AI make welfare decisions, $\beta = -0.19$, $t(1{,}137) = -4.27$, $p < .001$, 95% CI [−0.34, −0.04]. The average difference between the responses of claimants and non-claimants was 5.9 points (range: 0.3–12.8, see Fig. 3A).

Figure 3B displays the biases of claimants and non-claimants when trying to predict the answers of the other group, across the 36 trade-offs. Here we calculate the bias for each trade-off condition by subtracting participants' actual preference (e.g., claimants taking a claimant perspective) from the other groups' insights through perspective taking (e.g., non-claimants taking a claimant perspective). We then compare the bias scores with zero to determine their statistical significance, using the formula (1) below:

$$bias_{ij} = \beta_0 + \mu_{0j} + \epsilon_{ij} \tag{1}$$

where $bias_{ij}$ represents the bias for the $i$th observation in the $j$th participant, $\beta_0$ represents the fixed intercept, $\mu_{0j}$ represents the random effect for the $j$th participant, and $\epsilon_{ij}$ represents the residual error for the $i$th observation in the $j$th participant.

Both groups fail to completely take the perspective of the other group. On average, claimants underestimate the answers of non-claimants by 4.8 points, and non-claimants overestimate the answers of claimants by 6.4 points. Both biases are significantly different from zero, $\beta < 0.001$, $t(94) = -2.18$, $p = .032$, for claimants, and $\beta < 0.001$, $t(385) = 6.18$, $p < .001$, for non-claimants: the 95% confidence interval is [−9.2, −0.5] for claimants, and [4.4, 8.4] for non-claimants. Two issues when comparing the biases between the two groups, though, are their unequal size in our sample (20% as claimants and 80% as non-

claimants; the standard error for claimants is twice that for non-claimants), and the lack of financial incentives for responding correctly when taking the opposite perspective. These two issues are addressed in our second study.

In sum, data from our US representative sample shows that US citizens, on average, were willing to trade a 2.4 percentage point accuracy loss for a 1-week speed gain. However, welfare claimants are systematically more averse to AI than non-claimants, and we find evidence for a small asymmetry in the insights that claimants and non-claimants have into each other's answers: Non-claimants significantly overestimated claimants' preferences, while claimants significantly underestimated claimants' preferences.

### The UK balanced-sample study

To replicate the results obtained from the US representative sample, this study collected data from $N = 1462$ participants in the UK. Unlike the US representative-sample study, which had 20% claimants, we recruited an equivalent number of claimants and non-claimants in the UK. This sample size with a balanced composition of claimants and non-claimants can help consolidate our pre-registered hypothesis on the asymmetry in perspective-taking. In addition, we implemented the following changes:

1) We examined preferences about a specific benefit in the UK (the Universal Credit) and targeted a balanced sample between Universal Credit claimants (48%) and non-claimants (52%). The UK government recently announced the deployment of AI for the attribution of this benefit, raising concerns that the AI system may be biased against some claimants[7].

2) We adopted a different range of speed (0/1/2/3 weeks faster, as compared to a baseline waiting time of 4 weeks if handled by public servants) and accuracy (0/5/10/15/20% more false rejections than public servants) parameters, resulting in 20 trade-offs. Notably, when welfare AI demonstrates comparable performance (i.e., 0 week faster and 0% more error), people were still in favor of

## Perspective taking in the representative US sample

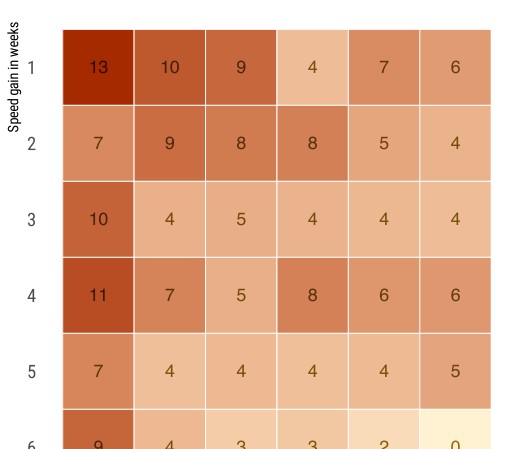

**(A)** Average gap in the willingness of claimants and non-claimants

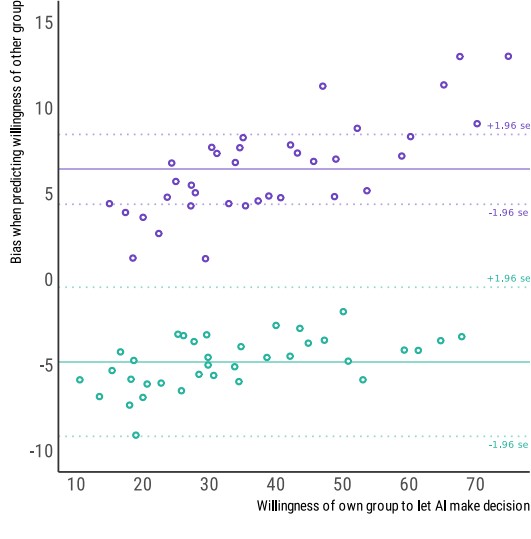

**(B)** Prediction bias, where each dot is one of the 36 tradeoffs

Size of gap · Welfare status · Not on welfare · On welfare

**Fig. 3 | Perspective taking in the US representative-sample study ($N = 987$; 20% as welfare claimants). A** Across all tradeoffs, non-claimants show greater willingness to let AI make decisions, a 13-point difference when speed gains and accuracy losses are low. **B** When trying to predict the answers of the other group, non-claimants overestimate willingness of claimants, and claimants underestimate the willingness of non-claimants.

humans making welfare decisions ($M = 45.4$, $SD = 28.7$; $t = -4.36$, $p < .001$, $d = -0.16$).

3) We added financial incentives for participants to correctly predict the preferences of the other group, that is, when non-claimants predict claimants' preference and claimants predict non-claimants' preference. We also asked non-claimants whether they had claimed welfare benefits in the past, whether they thought they may claim benefits in the future, and whether they were acquainted with people who were welfare claimants, to assess whether these circumstances made it easier to adopt the perspective of claimants.

4) For each trade-off, we additionally asked participants whether their trust in the government would decrease or increase (from 0 = decrease a lot to 100 = increase a lot) if the government decided to replace public servants with the AI program they just considered.

5) Finally, we added a treatment that made explicit the existence of a procedure to ask for redress in case a claimant felt their claim was unfairly rejected. Even though participants in the human redress (vs. no redress) condition believed in the chance to appeal in our manipulation check, $\beta = 0.37$, $t(3) = 55.80$, $p < 0.001$, 95% CI [0.36, 0.38], this clarification did not result in a statistically significant effect on trade-off preferences, $\beta = 0.03$, $t(737) = 1.26$, $p = 0.210$, 95% CI [−0.01, 0.07]. Therefore, we pool the data from this treatment with that of the baseline treatment.

Again, when participants responded from their own perspective ($N = 739$), their willingness to let AI make decisions was influenced both by speed gains, $\beta = 0.34$, $t(14,040) = 42.89$, $p < 0.001$, 95% CI [0.33, 0.35], and accuracy losses, $\beta = 0.44$, $t(14,040) = 57.51$, $p < .001$, 95% CI [0.43, 0.45]. Overall (see Fig. 4A), they traded off a 1-week speed gain for a 5 percentage point loss of accuracy. Among these UK participants, 47% self-declared as current claimants of the Universal Credit.

As in the US representative-sample study, for all 20 trade-offs, welfare claimants showed greater average aversion to letting AI make welfare decisions, $\beta = -0.09$, $t(735) = -2.64$, $p = 0.008$, 95% CI [−0.13, −0.05], with an average difference of 5.7 points (range: 0.1–8.7, see Fig. 5A). In both groups, we observed a correlation across trade-offs between the aversion to letting the AI make decisions, and the loss of trust in the government that would deploy this AI ($r = 0.77$ for claimants, and $r = 0.84$ for non-claimants). Moreover, for both groups, accuracy losses had a significantly stronger correlation with the loss of trust in the government ($r = 0.41$ for claimants, and $r = 0.42$ for non-claimants), compared to speed gains ($r = 0.21$ for claimants, $t(737) = 40.8$, $p < 0.001$, $q = 0.65$; $r = 0.25$ for non-claimants, $t(737) = 47.9$, $p < 0.001$, $q = 0.70$).

Figure 5B displays the biases of claimants and non-claimants when trying to predict the answers of the other group, across the 20 trade-offs. As in the US representative-sample study, we calculated the perspective-taking biases for claimants and non-claimants, respectively. On average, there is no statistically significant difference between claimants' estimates and non-claimants' responses, $\beta < -0.001$, $t(351) = -0.99$, $p = .323$, with an underestimation of 0.9 points and a 95% confidence interval including zero, [−2.7, 0.9]. Non-claimants, however, overestimate the preferences of claimants by 4.2 points, $\beta < 0.001$, $t(370) = 5.19$, $p < 0.001$, with a 95% confidence interval of [2.6, 5.7]. These asymmetrical insights between claimants and non-claimants are consistent with our preregistered prediction. To explore whether some life experiences may reduce bias in the predictions of non-claimants, we recorded whether they had past experience as claimants of other benefits, whether they were acquainted with current claimants, and their perceived likelihood of becoming claimants in the near future. We found no credible evidence for any of these effects.

In sum, results from the UK sample with a balanced composition of claimants and non-claimants consolidate and extend results from our US representative sample. The average willingness to trade

## Preferences for speed accuracy tradeoffs in the balanced UK sample

**(A)** Average preferences in the tradeoff conditions

**(B)** Preference distributions in the tradeoff conditions

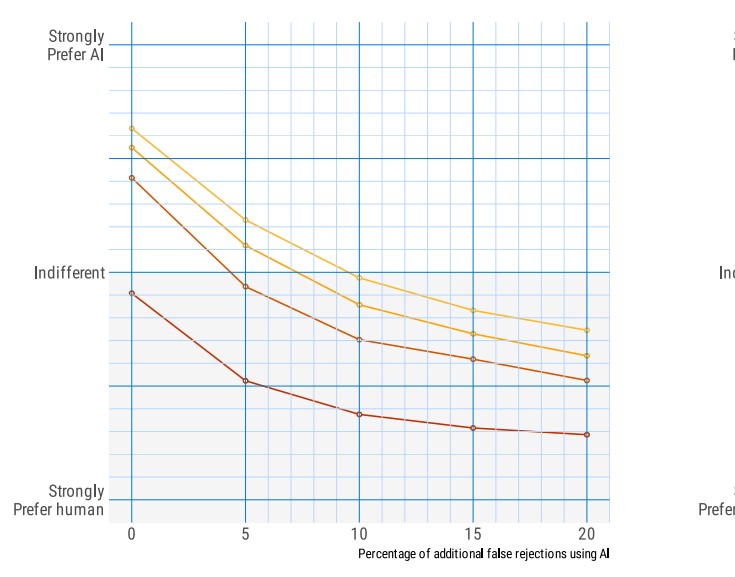
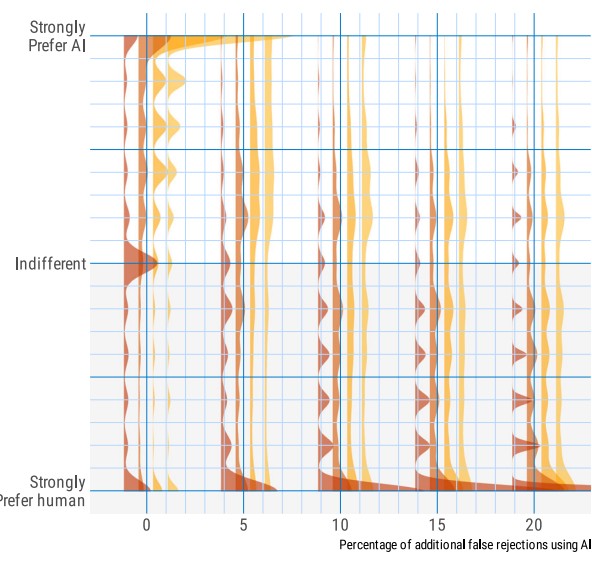

AI speed gain ⟶ 3 weeks faster ⟶ 2 weeks faster ⟶ 1 weeks faster ⟶ 0 week faster

**Fig. 4 | Preferences for speed accuracy trade-offs from own perspective, in the UK balanced-sample study (N = 739; 47% as welfare claimants). A** On average, respondents trade speed and accuracy at the value of 5 percent additional false rejections for speeding up the decision by 1 week. **B** Each slab shows the full distribution of individual preferences in each condition, grouped by speed gain.

## Perspective taking in the balanced UK sample

**(A)** Average gap in the willingness of claimants and non-claimants

**(B)** Prediction bias, where each dot is one of the 20 tradeoffs

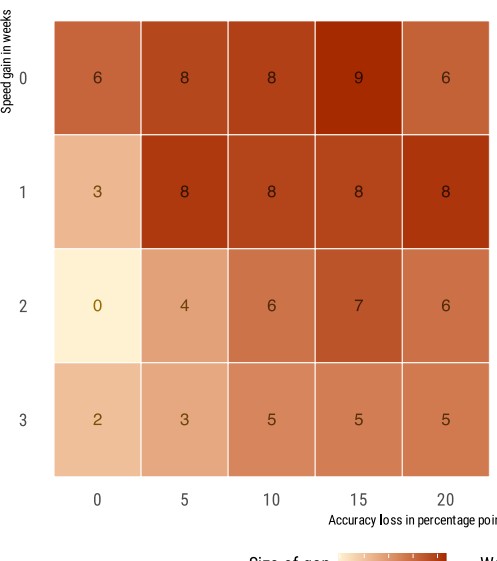
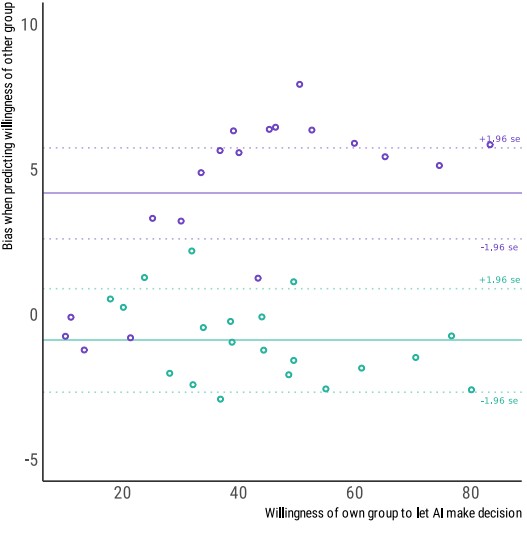

Size of gap | Welfare status ⟶ Not on welfare ⟶ On welfare

**Fig. 5 | Perspective taking in the UK balanced-sample study (N = 1462; 48% as welfare claimants). A** Across all tradeoffs, non-claimants show greater willingness to let AI make decisions, up to a 9-point difference. **B** When trying to predict the answers of the other group, non-claimants overestimate willingness of claimants, while claimants are close to providing unbiased estimates of the willingness of non-claimants.

a 5 percentage point accuracy loss for a 1-week speed gain hides heterogeneity in responses, with welfare claimants being systematically more averse to AI than non-claimants. We also find evidence for asymmetrical insights between claimants and non-claimants: Claimants' predictions do not differ significantly from the actual preferences of non-claimants, but non-claimants overestimate the willingness of claimants to let AI make decisions. Finally, lower acceptance of the AI system for welfare allocation is linked to decreased trust in the government among both welfare claimants and non-claimants.

### Main results of the US conjoint study

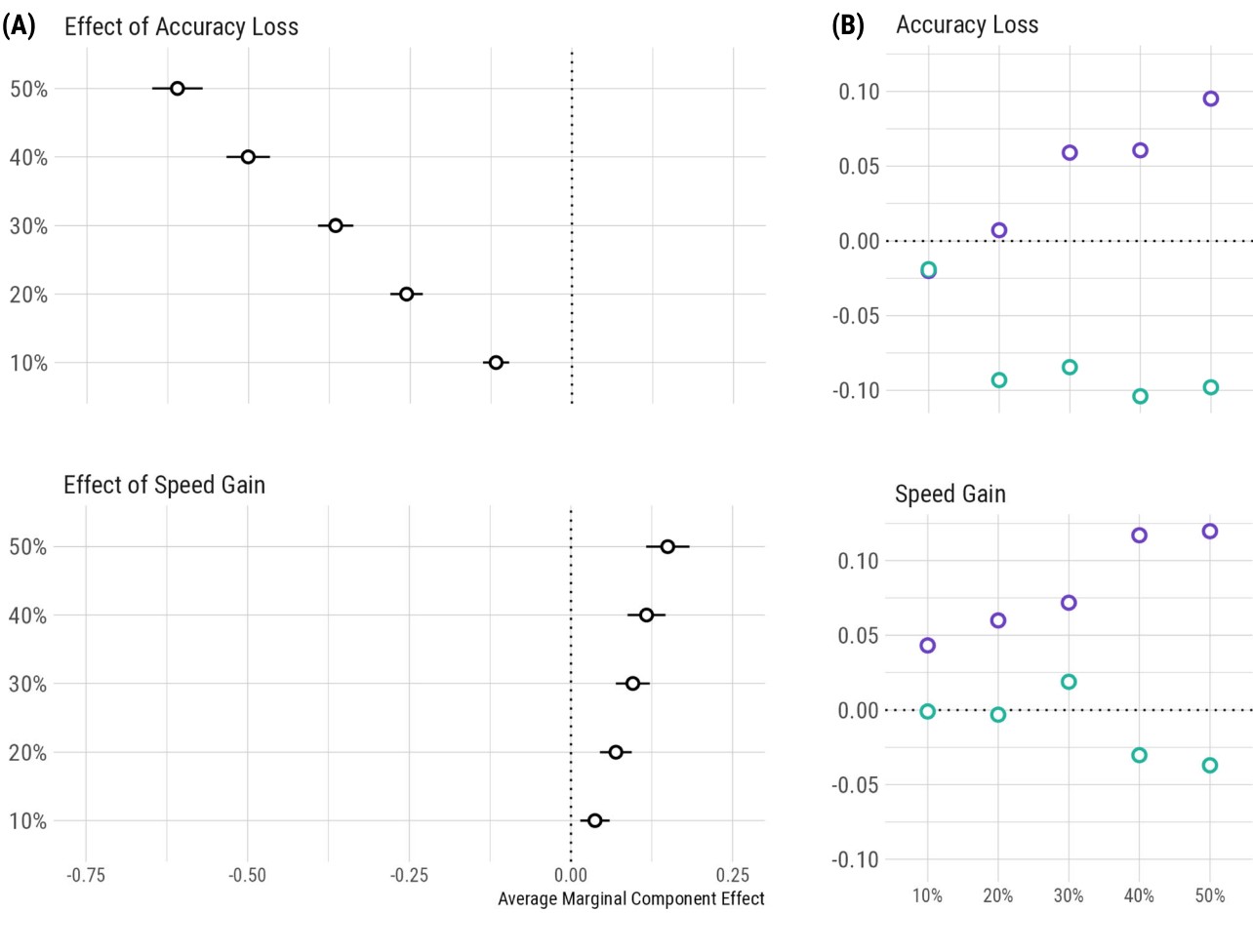

**Fig. 6 | Results of the US balanced-sample conjoint study. A** Average marginal component effects of each level of accuracy loss and speed gain on preferences for different AI systems (N = 402; 50% as welfare claimants): one percentage point of speed gain is equivalent to about 0.2 percentage points of accuracy loss. **B** Prediction bias when taking the perspective of others (N = 800; 50% as welfare claimants). Non-claimants are biased for both accuracy and speed, but claimants are only biased for accuracy.

### The US balanced-sample conjoint study

In the previous two studies, participants indicated preferences for individual AI programs, featuring speed gain by week and accuracy loss by percentage. This study aims to conceptually replicate previous findings in a choice-based conjoint experiment, where participants (1) select one of two AI programs presented in pairs and (2) evaluate the information of both speed gain and accuracy loss by percentage. We recruited a balanced sample of claimants and non-claimants from the US (N = 800). Each participant made binary choices for 30 pairs of AI programs, varying on speed gain (0%/10%/20%/30%/40%/50% shorter waiting time, as compared to a baseline of 40 working days if handled by public servants) and accuracy loss (0%/10%/20%/30%/40%/50% higher chance of false rejection, as compared to a baseline of 30% false rejection rate if handled by public servants). As such, we can infer participants' preferences for different AI programs, rather than human versus AI welfare decisions.

When participants responded from their own perspective (N = 402), their choices were influenced by both speed gains ($\beta$ = 0.24, z = 8.91, p < 0.001, 95% CI [0.21, 0.27]) and accuracy losses ($\beta$ = 0.96, z = 35.09, p < 0.001, 95% CI [0.93, 0.99]) of different welfare AI programs. We used conjoint analysis to compute the average marginal component effect (AMCE) at each attribute level, relative to 0% speed

gain and 0% accuracy loss, respectively (see Fig. 6A). On average, an AI speed gain by 1 percentage point increases the probability of choice by 0.2% (SE = 0.0002), and an AI accuracy loss by 1 percentage point reduces the probability of choice by 1.1% (SE = 0.0001). Put differently, people were willing to tolerate a 0.2 percentage point of AI accuracy loss for each 1 percentage point increase in speed. Among these US participants, 50% self-declared as welfare claimants. Overall, for each 1 percentage point increase in speed, claimants and non-claimants were willing to tolerate a 0.2 and 0.3 percentage point of AI accuracy loss, respectively. As shown in Table 1, in three out of five non-zero accuracy loss levels (30%/40%/50% more false rejections), AMCEs were more negative for non-claimants than claimants (z > 2.03, ps < 0.05). However, the relative difference in accuracy loss is small, totaling a 9.1% smaller weight for claimants. In contrast, the relative difference in speed gain is large, amounting to a 50.6% smaller weight for claimants. Across all five levels of non-zero speed gain, AMCEs were more positive for non-claimants (z > 2.35, ps < 0.02). To summarize, claimants and non-claimants put relatively similar importance on accuracy losses, but claimants put lower importance on speed gains.

Figure 6B displays the biases on speed and accuracy, respectively, when claimants and non-claimants make choices from each other's perspective. Overall, when taking the other group's perspective,

**Table 1 | Comparisons of average marginal component effects (AMCEs) between claimants and non-claimants at each level of accuracy loss and speed gain**

| | Accuracy Loss | | | | Speed Gain | | | |
|---|---|---|---|---|---|---|---|---|
| | Claimants [95% CI] | Non-claimants [95% CI] | z (p) | d | Claimants [95% CI] | Non-claimants [95% CI] | z (p) | d |
| 10% | −0.12 [−0.15, −0.10] | −0.11 [−0.14, −0.09] | −0.55 (.585) | −0.77 | 0.01 [−0.01, 0.04] | 0.06 [0.03, 0.09] | −2.36 (0.018) | −3.33 |
| 20% | −0.25 [−0.28,−0.22] | −0.26 [−0.29, −0.24] | 0.76 (.449) | 1.06 | 0.05 [0.02, 0.08] | 0.09 [0.06, 0.12] | −1.97 (0.049) | −2.78 |
| 30% | −0.35 [−0.37, −0.32] | −0.39 [−0.41, −0.36] | 2.04 (.042) | 2.88 | 0.07 [0.04, 0.10] | 0.12 [0.10, 0.15] | −2.74 (0.006) | −3.88 |
| 40% | −0.48 [−0.50, −0.45] | −0.53 [−0.55, −0.50] | 2.70 (.007) | 3.82 | 0.07 [0.04, 0.10] | 0.16 [0.14, 0.19] | −4.71 (< 0.001) | −6.66 |
| 50% | −0.58 [−0.61, −0.56] | −0.64 [−0.66, −0.62] | 3.30 (< 0.001) | 4.66 | 0.11 [0.08, 0.14] | 0.19 [0.16, 0.22] | −4.09 (< 0.001) | −5.78 |

claimants and non-claimants were willing to tolerate a 0.3 and 0.4 percentage point of AI accuracy loss, respectively, for each 1 percentage point increase in speed. Conceptually similar to the two previous studies, we calculate the biases by subtracting the targeted group's AMCEs from the other group's AMCEs through perspective-taking. On average, claimants underestimate the importance of accuracy for non-claimants by 8.0 percentage points, 95% confidence interval [0.04, 0.12], $t(4) = 5.14$, $p = 0.007$, $d = 2.30$, but showed no statistically significant bias in estimating the importance non-claimants placed on speed, $t(4) = 2.55$, $p = 0.064$, $d = 1.14$, with an underestimation of 1.8 percentage points and a 95% confidence interval including zero, [−0.002, 0.04]. In contrast, non-claimants overestimate the importance of both speed and accuracy for claimants, by 8.2 points (95% confidence interval [0.04, 0.12]), $t(4) = 5.29$, $p = 0.006$, $d = 2.36$, and 4.9 percentage points (95% confidence interval [0.005, 0.09]), $t(4) = 3.08$, $p = 0.037$, $d = 1.38$, respectively. These findings again corroborate the asymmetrical insights between claimants and non-claimants into each other's perspective.

In summary, our conjoint study reveals that US citizens, comprising a balanced sample of claimants and non-claimants, are willing to trade off a 0.2 percentage point reduction in AI accuracy for each 1 percentage point increase in speed. Beneath this overall pattern, however, non-claimants respond more positively than claimants to speed gains of welfare AI programs. We further show the asymmetry in perspective-taking: while claimants accurately predict the importance of speed for non-claimants, non-claimants overestimate the importance of both speed and accuracy for claimants. These findings conceptually replicate previous studies, and suggest that claimants may prioritize other factors that non-claimants fail to recognize. By employing a different design and analysis strategy, the conjoint study further strengthens the robustness of previous results, demonstrating that they are not tied to a specific experimental setup.

## Discussion

One primary advantage of using AI for welfare benefit allocation is quicker decision-making, allowing claimants to receive support faster[1,5,6]. However, these systems often result in an accuracy loss, potentially leading to unfair denials or false fraud accusations[5–10]. Governments deploying welfare AI systems may need to navigate these trade-offs carefully, particularly given their potential impacts on public trust[18,31]. Our findings also suggest that the acceptability of these trade-off decisions is correlated with public trust in the government.

Collecting data from the US and UK ($N = 3249$), our study suggested that participants would trade a one-week speed gain for a 2.5 to 5 percentage point accuracy loss, or 1 percentage point speed gain for 0.2 percentage point accuracy loss. However, we also found that averaging across participants masked divergences between claimants and non-claimants. Though the difference between the two groups

varied across trade-offs, welfare claimants were systematically less amenable to AI deployment than non-claimants. This finding aligns with recent calls in behavioral science to focus on heterogeneity when informing policy[32], as well as to consider the positionality of AI models[33], that is, their social and cultural position with regard to the stakeholders with which they interface. In summary, average responses may not capture the divergent preferences of stakeholders in welfare AI systems. It is easy to imagine that efficiency gains – such as a more cost-effective government and increased labor availability – could be enough to convince the majority, non-claimant population to accept welfare AI systems and improve their trust. However, if governments aim to align welfare AI systems with claimants' preferences, they may need to look beyond aggregate public opinion, as it does not necessarily capture the perspectives of those directly affected.

Data revealed a further complication: asymmetric insights between claimants and non-claimants. While neither group was perfectly accurate in understanding the perspective of the other, non-claimants were more likely to provide biased estimates of claimant's preferences or choices, even in the presence of financial incentives. These findings echo laboratory results suggesting that participants who are or feel more powerful struggle to take the cognitive perspective of others[34–37], as well as sociological theories positing that marginalized groups have greater opportunities and motivations to develop an understanding of the thoughts and norms of dominant groups[38–40]. In the context of welfare AI, asymmetric insights create the risk that the perspective of claimants may be silenced even when non-claimants seek to defend the interests of claimants. These well-intentioned non-claimants may use their dominant voice to shape public opinion and policy without realizing that they do not in fact understand the preferences of claimants, resulting in AI systems that are misaligned with the preferences of their primary, direct stakeholders. Our results thus underline the need to involve potential claimants in the co-design process, or develop technical solutions that incorporate their perspectives and preferences when configuring AI in welfare systems – rather than to assume that their preferences are well-understood or can be understood through empathetic perspective-taking.

Our results also shed light on the potential for transparent communication about the performance and alignment choices of welfare AI systems, especially in the political decision-making processes involving the general public. First, we demonstrate that people can systematically evaluate the benefits and costs of deploying welfare AI systems, rather than focusing solely on negative features. Public disclosure of AI inaccuracies does not simply lead to criticism and pushback; people also value the accompanying speed gains and relative accuracy improvements, possibly over time. Second, we offer scientific support for public communication strategies when welfare AI systems prioritize the preferences of a small subgroup of claimants

over the majority, non-claimant population. These decisions about whose values and preferences AI aligns with – often referred to as 'the alignment problem'[41–43] – can be justified by the realities of heterogeneous preferences and asymmetrical insights in the context of welfare decisions.

This research is, however, limited in at least two important aspects. First, while we identified correlations between tradeoff preferences and trust in government, we did not directly test how public trust evolves with full versus partial transparency. The real-life mechanisms underlying public trust are more complicated than this study addressed[44,45]. For example, beyond information from governmental agencies, public opinion is increasingly influenced by exposure to the suffering of vulnerable individuals due to algorithmic mistakes on social media and news platforms[7,10]. Future research should systematically examine how trust in government changes when different aspects (e.g., technical, anecdotal, societal) of welfare AI systems are communicated.

Second, the current research focused on one critical type of tradeoff for the deployment of AI in welfare systems: the speed-accuracy tradeoff. We operationalized AI inaccuracy as additional false rejections compared to human conditions, and found that claimants were more averse than non-claimants to welfare AI programs and their mistakes. However, these findings do not imply that AI programs should not be launched before they become perfect. They also do not suggest that human decisions are error-free, or that AI always makes more mistakes than humans in welfare decisions. In reality, AI- and human-dominant government systems may face different challenges. For example, AI can be hyper-vigilant about anomalies[8,9] and seen as inflexible in self-corrections[18,19]. In contrast, human public servants may discriminate against particular social groups, and such biased judgments may vary from person to person and induce inconsistencies and unfairness in welfare payments[46,47]. Therefore, future research may explore public opinions for other tradeoffs, such as different types of inaccuracy introduced by AI versus human welfare systems.

More broadly, despite increasing technical attempts to align AI with pluralistic values and diverse perspectives[41–43], there are inevitably situations where agreement or reconciliation cannot be easily achieved (e.g., when non-claimants fail to estimate welfare claimants' aversion to AI, but not vice versa). Our core findings, heterogeneous preferences and asymmetric insights, may also hold in other cases where AI is deployed in a context of power imbalance – conducting behavioral research on these cases in advance of AI deployment may help avoid the scandals that marred the deployment of welfare AI.

## Methods

All three studies complied with relevant ethical regulations for human subjects, were approved by the ethics committee at the Max Planck Institute for Human Development (NO. A2022-01, A2022-18, and A2024-16), and obtained informed consent from all participants. Data were collected in February 2022, September 2022, and December 2024, respectively. Participants' sex was determined based on self-report. Sex was considered in the study design to ensure demographic representativeness or a sex-balanced sample. Data deaggregated by sex are provided in the Supplementary Notes B.1 to B.3, with consent obtained for reporting and sharing anonymized individual-level data. All participants were recruited on Prolific for a study named "Artificial Intelligence in Social Welfare". Upon completion, participants in the first two studies were paid £1.6, and participants in the conjoint study were paid £1.8. Participants in the UK balanced-sample study who had to predict the answers of the other group (but not their own group) received an additional £0.03 for each response that fell within 5 points of this other group's average.

All studies were programmed and hosted on Qualtrics. After providing informed consent and basic demographic information,

participants were randomly assigned to take a claimant or non-claimant perspective, while the investigators were not blinded to allocation during experiments and outcome assessment. To familiarize participants with the stimuli and response scale, they were first shown two exercise trials in the survey. In the first two studies, the exercise trials each presented one extreme speed-accuracy combination, while the last conjoint study showed two such cases side by side. Participants completed these two trials and had a chance to review and change their answers. Then the survey started, and all targeted speed-accuracy combinations were shown in random order. The style of presenting speed and accuracy information closely follows relevant public-facing government reports. Complete descriptions of our materials and survey questions are included in the Supplementary Methods A.1 to A.3.

All statistical analyses were conducted using R (version 4.3.1) within the RStudio environment (version 2024.09.1). The following R packages were used: tidyverse[48] (version 2.0.0), lme4[49] (version 1.1.35.5), lmerTest[50] (version 3.1.3), effectsize[51] (version 1.0.0), margins[52] (version 0.3.28). We assessed normality of residuals and homogeneity of variances where appropriate. Unless otherwise stated in the results, no major violations of assumptions were detected. For all parametric tests and regression models, two-tailed tests were applied.

### The US representative-sample study

**Participants.** We had $N = 987$ participants from the United States, who were representative on age ($M = 45.3$, $SD = 16.3$), sex (473 males and 514 females), and ethnicity (77.8% White, 11.4% Black, 6.1% Asian, 2.5% Mixed, and 2.1% other), and 20.4% of them self-reported as welfare claimants at the time of the study; no data were excluded from the analyses. The sample size was determined based on the recent recommendation of around 500 people for latent profile analysis[53]. We aimed for an almost doubled sample size given our two-condition perspective-taking manipulation.

**Design and procedure.** The US representative-sample study employed a mixed design, with one between-subjects and two within-subjects factors. First, participants were randomly assigned to take a claimant ("You are applying for a social benefit") or a controlled tax-payer ("Someone else in your city is applying for a social benefit") perspective. We then manipulated the information about welfare AI's speed (6 conditions: 1/2/3/4/5/6 weeks faster than a public servant) and accuracy (6 conditions: 5/10/15/20/25/30% more false rejections than a public servant). The presented speed (an average of 8 weeks) and accuracy (at most 30% more errors) baselines referred to realistic information from some governmental reports and third-party investigations[9,28–30].

After knowing the perspective they should take, participants went through two exercise trials, reading two extreme cases of welfare AI (bad case: 0 week faster + 50% more false rejections; good case: 7 weeks faster + 1% more false rejections) and answering the same question "To what extent do you prefer a public servant or the AI program to handle your/the person's welfare application?" (from 0 = definitely a public servant to 100 = definitely the AI program). These exercise trials aim to familiarize participants with the experiment paradigm. Therefore, we gave participants a chance to review the example stimuli and their corresponding answers, and calibrate their answers before moving to the 36 official test rounds. The official test rounds no longer allowed revisions, including going back to previous pages or revising validated answers to previous tradeoff scenarios, which was clearly explained to participants at the end of the training session. In each of the 36 test rounds, they read information about their perspective, AI speed, and AI accuracy in three consecutive cards (see Supplementary Fig. 1 for an illustration of the cards in different experiment conditions). After reading the three cards in each round, participants answered the same question about their preference for welfare AI versus public servants.

## The UK balanced-sample study

**Participants.** We performed a simulation-based power analysis for multilevel regression models, which suggested that a sample of $N = 800$ would allow us to detect the interaction effect of AI performance, claimant status, and perspective-taking with higher than 80% power at an alpha level of 0.05. We therefore aimed for $N = 1600$ participants in the United Kingdom given our additional between-subjects human redress manipulation. The study was preregistered on Open Science Framework on September 27, 2022, which can be assessed at https://doi.org/10.17605/OSF.IO/54PX2. All procedures and analyses were conducted as in the preregistered protocol. As preregistered, we filtered out participants who provided different answers to one identical welfare status question ("Are you a recipient of Universal Credit?"; Answer: "Yes/No"), which was embedded both in the Prolific system screener and our own survey. After the screening, no other data were excluded from the analyses. We eventually had $N = 1462$ participants (age: $M = 37.6$, $SD = 11.1$; ethnicity: 88.4% White, 3.0% Black, 5.6% Asian, 2.7% Mixed, and 0.3% other), with a relatively balanced composition of males and females (42.7% male, 55.9% female, 1.4% other), and welf are claimants (47.9%) versus non-claimants (52.1%).

**Design and procedure.** The balanced-sample study examined a real-life social benefits scheme in the UK – Universal Credit (https://www.gov.uk/universal-credit). We employed a mixed design with three between-subjects and two within-subjects factors. As between-subjects factors, we recruited both Universal Credit claimants and non-claimants, and randomly assigned them to take a Universal Credit claimant or a controlled taxpayer perspective. They were then randomly assigned to a no redress or a human redress condition, which differed on whether claimants could appeal to public servants. As within-subject factors, we manipulated information about welfare AI's speed (0/1/2/3 weeks faster, as compared to a baseline waiting time of 4 weeks if handled by public servants), and accuracy (0/5/10/15/20% more false rejection).

Before starting the 20 rounds of official tradeoff evaluations, participants again went through two exercise trials with a chance of revision, reading two extreme cases of welfare AI (bad case: 0 week faster + 40% more false rejections; good case: 3 weeks faster + 1% more false rejections). In each example, they answered two questions: "To what extent do you prefer a public servant or the AI program to handle your/the person's welfare application?" (0 = definitely a public servant to 100 = definitely the AI program) and "If the UK government decided to replace some public servants with the AI program in handling welfare applications, would your trust in the government decrease or increase?" (0 = decrease a lot to 100 = increase a lot). They then had a chance to review and change their answers to the two cases. After moving to the 20 official test rounds, they were no longer allowed to go back to previous pages or revise answers to previous tradeoff scenarios, which was clearly explained to participants at the end of the training session. In each of the 20 test rounds, they read information about their perspective, AI speed, AI accuracy, and human redress condition in three consecutive cards (see Supplementary Fig. 2 for an illustration of the cards in different experiment conditions). After reading the three cards in each round, participants answered the same two questions about their preference for welfare AI versus public servants, and their trust in the government.

To increase the motivation of perspective taking, participants were informed and incentivized to take the opposite perspective, for each accurate answer that fell within ±5 points of the other group's average. At the end of the 20 official rounds, as a manipulation check, participants indicated the extent to which they believed that "you/the person can appeal to public servants if you/they are not satisfied with the welfare decision made by the AI program?" (0 = not at all to 100 = very much).

We also pre-registered three predictions. First, we expected that accuracy losses would matter more to participants than speed gains. Practically speaking, we expected that participants would value one experimental unit of speed gains (1 week) less than one experimental unit of accuracy loss (5 percentage points). This prediction was not supported since participants tolerated a 5 percentage point accuracy loss for a 1-week speed gain. Second, we expected to identify subgroups of participants with different patterns of trade-offs between speed gains and accuracy losses. This prediction lacked support since we did not find a particular number of latent profiles that significantly outperformed others (see the Supplementary Notes B.2). Third, we expected that non-claimants would not be good at predicting the preferences of claimants. The results were consistent with the third prediction.

## The US balanced-sample conjoint study

**Participants.** Our a-priori power analysis using the cjpowR package (version 1.0.0)[54] suggested a minimum sample of $N = 157$ for our conjoint experiment design (i.e., two profiles, each with six levels, and 30 trials for each participant; with 80% power at an alpha level of 0.05). We then aimed for 200 participants in each of the four claimant status by perspective-taking conditions. The final $N = 800$ participants (age: $M = 40.9$, $SD = 14.2$; ethnicity: 59.5% White, 29.8% Black, 4.5% Asian, 4.0% Mixed, and 2.2% other) had a roughly balanced composition of males and females (53.0% male, 46.2% female, 0.5% other, and 0.2% prefer not to say). Through a prescreen survey released earlier on the same day, we were able to release the main study to a balanced sample of welfare claimants (50.3%) versus non-claimants (49.7%); no data were excluded from the analyses.

**Design and procedure.** The conjoint study employed a mixed design, with one between-subjects and two within-subjects factors. Participants were randomly assigned to take a claimant or taxpayer perspective. We then manipulated the information about welfare AI's speed (6 conditions: 0/10/20/30/40/50% shorter waiting time, as compared to a baseline of 40 working days if handled by public servants) and accuracy (6 conditions: 0/10/20/30/40/50% higher chance of false rejection, as compared to a baseline of 30% false rejection rate if handled by public servants). It was challenging to determine a realistic human baseline across various welfare AI programs. We therefore relied on a recent public opinion study[55] and set the human false rejection rate at 30% to align with common estimations.

After knowing the perspective they should take, participants went through two exercise trials, each presenting a pair of welfare AI programs side by side (0% faster and 50% more false rejections, versus 50% faster and 0% more false rejections; 0% faster and 0% more false rejections, versus 50% faster and 50% more false rejections). They answered the same question "Which AI program would you prefer?" by selecting one of the two AI programs. Before moving to the 30 official test rounds, they had a chance to review and calibrate their answers in the exercise trials. They were reminded again of the human baseline conditions (40 working days and a 30% rate of false rejection) and were informed that the test rounds would no longer allow revisions. In addition to speed gain and accuracy loss information by percentage, participants also read information about the actual waiting time and false rejection rate of the AI program. For example, corresponding to an AI program being 10% faster and having 10% more false rejections, we noted that "10% shorter waiting time = 36 working days; 10% higher chance of false rejection = 33% false rejections" (see Supplementary Fig. 3 for an illustration of the cards in different experiment conditions).

## Reporting summary

Further information on research design is available in the Nature Portfolio Reporting Summary linked to this article.

## Data availability

The raw datasets[56] generated and analyzed during the current study are available in the Open Science Framework repository, https://doi.org/10.17605/OSF.IO/Z637M.

## Code availability

All code[56] necessary to reproduce all analyses is openly accessible in the Open Science Framework repository, https://doi.org/10.17605/OSF.IO/Z637M.

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

## Acknowledgements
We acknowledges support from grant ANR-19-PI3A-0004 [J.F.B.], grant ANR-17-EURE-0010 [J.F.B.] and the research foundation TSE-Partnership [J.F.B.].

## Author contributions
Conceptualization: M.D., J.F.B., I.R.; Methodology: M.D., J.F.B.; Investigation: M.D.; Visualization: M.D., J.F.; Funding acquisition: I.R.; Project administration: M.D.; Writing – original draft: M.D., J.F.B.; Writing–review & editing: M.D., J.F.B., I.R.; Materials & Correspondence: Correspondence and requests for materials should be addressed to M.D.

## Funding

## Competing interests
The authors declare no competing interests.
