## [Transparent Peer Review File · Nature Communications]

Heterogeneous preferences and asymmetric insights for AI use among welfare claimants and non-claimants

Corresponding Author: Dr Mengchen Dong

Version 0:

Reviewer comments:

Reviewer #1

(Remarks to the Author)

This is a very interesting and timely paper on the perceptions of the use of AI in social welfare systems. The authors use two large samples (one from the US and the other from the UK, the latter pre-registered) to study the acceptance of trade-offs between speed and accuracy in the use of AI in the welfare processing system. The whole sample is, on average, willing to trade off speed at a loss of accuracy. However, subgroups including claimants and vulnerable populations are less inclined to exhibit such a trade-off and generally oppose the use of AI, especially as accuracy declines. This may seem unsurprising in hindsight, but one could have also imagined that the improved efficiency of AI (even at a slight cost of mistakes) could appeal to an audience in need for such benefits quickly.

This paper is well written, the methods are solid, the analysis is robust, and the results are both of general interest and nuanced in important ways. I like the figures in particular, they clearly articulate the findings, also to a broader audience.

This paper will provide impetus for other scholars to work on these important questions in the future (including with understudied populations such as claimants), but it also has immediate implications for policymakers and the welfare sector – this is critical since we are at a time when we need timely and robust research to inform policy-making around AI.

I have the following comments and suggestions for the authors:

1) The asymmetry between claimants' and non-claimants' perspective taking (or lack thereof) is interesting. Claimants, due to their vulnerable status, are an understudied population and I'm glad that you made the effort to look into this population's perspective (as, obviously, they would be most affected by the use of AI in this domain).

1a) When you introduce the UK study (end of page 2), it may be worth pointing out as another "difference" to the US study that you recruited a targeted sample of claimants and non-claimants. Because it is so intentional, I believe it is worth mentioning as a deliberate methodological change to your design in this study and the sample size to reach meaningful conclusions.

1b) In general, I would appreciate having the sample sizes for each panel (and each subgroup, e.g. claimants and non-claimants, where appropriate), either in the figure legends or inside the graphs.

2) A minor observation from your results in Study 2 might be worth drawing out: when AI has no accuracy loss (i.e. 0 on the x-axis) but there are also no speed gains (0 weeks faster), the whole sample seems to be on average "indifferent" between human and AI (and, line with earlier results, group 3 still prefers human across the board). This may seem unimportant at the surface but from a public policy position (and taxpayer perspective!), it would be so much more cost-effective to use AI over human – in this case without the real benefits or drawbacks of AI, except that, once running, the AI saves a lot of labour costs. Of course, we are a long way away from a "perfect error-free AI", but policy-makers and scholars alike might be interested in seeing that there is less of an "algorithm aversion" than has previously been documented.

3) In the discussion section, I think it would be good to make explicit a counterfactual not currently discussed: assume you don't use AI and just rely on humans – they, too, make errors. So, what are the welfare gains from using AI that avoid the

opposite risk, i.e. that human decision-makers in the absence of AI would make an inaccurate decision – but without benefit of speed! Put differently, it is worth remembering for scholars and policy-makers alike that neither humans nor AI are infallible, so future research may want to take a closer look at what people would prefer if they could only choose between fallible AI and human options.

4) Personally, I'm less convinced of the value of the pre-registration given that you already have two sizeable samples from two different countries and, importantly, find largely similar results: given that most of your estimates with the same analyses are pretty consistent across the two studies (except the difference of the accuracy of perceptions of non-claimants between the two studies), I think the pre-registration isn't really telling us much that we didn't already expect to see from pooled $n=2,500$ and a conceptual replication. However, I appreciate the transparency in including the directional hypotheses you made for study 2 as well as the fact that some of those hypotheses did not match the empirical findings in study 2. That said, I think not having these predictions in the main text would improve readability (at no loss of accuracy!). (And these pre-registered hypotheses could just as well be mentioned in the Materials and Methods section.) But this is a matter of preference.

(Remarks on code availability)

I have looked at the accompanying HTML files, which show the code and the output. The code and outputs look clear and readable.

Reviewer #2

(Remarks to the Author)

Key results

The study's key result is showcasing the heterogeneity between claimants of welfare and non-claimants of welfare in how acceptable they find decreases in accuracy for a faster welfare service when replacing decision-making with AI. Welfare claimants were far less willing to trade off speed for accuracy. Additionally, non-welfare claimants were found less capable of taking claimants' perspectives. The authors combined this with the fact that non-claimants make up the majority to conclude that by averaging opinions in a majority rule system, we – as a society – may falsely believe a certain AI implementation is desirable, despite chief stakeholders potentially disagreeing.

General feedback

Overall, the study is technically well done and provides some interesting results. The findings are documented well and pleasantly written up. As far as we are aware the contribution is novel. However interesting, the overall significance of the results for a general audience – as could be expected for this journal – seems somewhat narrow. In addition, there may also be some issues with the design of the experiment. We will explain these points below, in addition to some more specific points of feedback.

- Line 52: We find the phrasing of “this balance must be informed by public preference” somewhat questionable. The assumption that majority preference results in changes in AI design biased against vulnerable groups seems overly simplistic and not grounded in current theory. While it may be true that vulnerable stakeholders may be missed out on in AI design, there are more complicated mechanisms at work that are overlooked in this paper. First, if majority preferences have an effect this is very indirectly: e.g., this may be caused by political decision-making that limits the autonomy of welfare case workers to grant welfare, but this often requires legal changes which only indirectly leads to AI design changes. Second, when developing AI systems, managers and developers tend to have a technical-rational perspective and thereby overemphasizing accuracy and efficiency while missing out on taking stakeholder perspectives at large (For this and our other points on value-sensitivity and how this is translated into practice, see for example Fest, I., Schäfer, M., van Dijck, J., & Meijer, A. (2023). Understanding Data Professionals in the Police: A Qualitative Study of System-Level Bureaucrats. *Public Management Review*, 0(0), 1–21. <https://doi.org/10.1080/14719037.2023.2222734>).
- Line 210-211: “In summary...of stakeholders”. We think this summation states a stronger conclusion than can be inferred from the article. The article convincingly shows how following popular opinion can result in the main stakeholders of a deployed AI system not seeing their design desires reflected, so perhaps such a system fails in giving voice to those it aims to help. However, even if design specifications are inspired by popular opinion, the study does not show that this results in a degradation of government trust at large. It is easy to imagine that the efficiency gains implied by the article result in e.g. a cheaper government and more labour available elsewhere, which may be a sufficient reason for the majority to still prefer to push through their opinion and improve their trust.
- Line 221-227: This paragraph seems contradictory. Non-claimants are unable to take the perspective of claimants even when they try, so the solution is to actively engage with claimants. However, presumably, those engaging with them are predominantly non-claimants, so how does this solve anything? I would suggest putting these vulnerable stakeholders in a position of co-design or letting subsequent research find ways to allow their perspectives to be taken.
- Line 228: The argument for transparent communication is still underdeveloped. Several questions remain unanswered, e.g. Does this solve the issue addressed by this study, and how? What is the target audience of the communication and what would the communication entail?

Data, methodology & validity

Overall, the methodological approach is well documented. The data is freely available and made very accessible through the provided description. For the most part, applied methodology is sound and should provide some level of validity to real-world contexts. Yet we also have the following more specific points:

- We have already brought up our concerns over the external validity of this study in the general section. Specifically, we question both whether the popular opinion is translated into design specifications of AI or other political and administrative decisions, as well as whether a speed/accuracy tradeoff is currently (widely) used as a pre-specified design consideration for public sector AI use.
- One important concern regards the vignette design, and in particular, how speed and accuracy are presented to

participants. Speed is presented with a baseline and improvements are an absolute amount (e.g. it takes 4 weeks, AI makes it 3 weeks faster). However, the accuracy improvement is presented as a relative percentage without a baseline (AI helps detect fraud and leads to a 20% higher chance of rejection). First off, this means that if claimants and non-claimants are simply divergent in the baseline they assume/experience in existing rejection, you are measuring that difference. For example, if claimants already experience a 50% false rejection rate, then any decrease in accuracy is very undesirable. However, if non-claimants assume a 1% false rejection rate, even a 100% higher false rejection rate is somewhat fine if it speeds up the process. We would argue that you would require some knowledge of what people across these groups think the baseline accuracy is to conclude. This can then still be a conclusion about divergence in participants taking one another's perspective, though the reason may be rooted in a lack of knowledge instead of an inability to take perspective. Secondly, people in general are not great at dealing with estimates in the way you present them. Even if the baseline of let's say 10% rejection is assumed by everyone and we are given a 2-week speed boost, a 20% increase in false rejection seems like a very significant increase with 2 weeks not sounding like that much at all, but in absolute terms this would only result in the total rejection being 12% or a 2% difference in absolute terms while it would be a 50% increase in speed. Just by phrasing it in terms consistent with one another, we would argue the trade-off sounds much more reasonable without changing at all (For both these points, work by the behavioural economist Gigerenzer such as "The bias bias in behavioural economics" could provide some additional context).

- Line 272: It was a bit unclear as to why revision of answers in the survey was not allowed anymore. Perhaps provide a reason for this.
- Line 310-314: There are methodological discussions about how well financial incentives work. Perhaps those less well-off (the claimants) are more motivated to answer correctly (or even seek out information to allow them to answer correctly).

Suggested improvements

Mostly of the following points reiterate our comments above, but the following are some of our chief suggestions for improvement and extension:

- A more nuanced argumentation of how the preferences of stakeholders end up inspiring AI design, as one may, for example, find in work on value-sensitive design.
- A more intricate design. For example, perhaps having the option to appeal to a human decision-maker might make it very worthwhile to accept a very fast but inaccurate AI system, which could be combined with how discernible it is that you are falsely and not rightly rejected.
- An extension regarding perspectivetaking which performs a comparison of speed and accuracy o absolute terms with the presence of a baseline
- An extension that shows the discussed issue is based on lack of knowledge (claimants know the inaccuracy is higher than the non-claimants think it is), perspective taking (non-claimants are unable/unwilling to empathize with the position of claimants though they share knowledge) or perhaps both.
- Why only consider the speed/accuracy tradeoff in the chosen direction? Perhaps claimants would first want both faster and more accurate systems before it is worthwhile, suggesting support for AI introduction in this group is dependent on other factors. Similarly, perhaps claimants are happy to take a slow-down for higher accuracy. This could be a worthwhile tradeoff for example if normally you are rejected so often that a slightly slower system is worth it. The preceding may not be a realistic scenario for AI introduction, but could steer policymakers to prioritize accuracy at the cost of speed.

References:

For the most part, the provided references are appropriate. Beyond what was mentioned in the general section, we have the following points:

- Line 45 & line 199: As far as we can tell, he references to sources 4 and 5 seems out of place. Neither article referenced is about the preceding statement concerning the main promise of introducing AI in decision-making for welfare supposedly speeding up decision-making. At best, both articles make some mention of the efficiency and effectiveness of AI in government at large or refugee placement specifically. Yet for a generic point on efficiency & effectiveness concerning AI in the public sector, other sources would probably be more appropriate. Neither of the provided sources specifically covers speeding up decision-making through AI or this being considered the main promise for its introduction in welfare.

(Remarks on code availability)

Reviewer #3

(Remarks to the Author)

(Remarks on code availability)

Reviewer #4

(Remarks to the Author)

The authors explore preferences for speed-accuracy tradeoffs between people who do and do not claim social benefits, with the aggregate sample in the US study reflecting the census data (i.e., nationally representative).

The key finding of the paper is that non-claimants overestimate the degree to which claimants prioritize convenience over accuracy. This illustrates how nationally representative samples may fail to accurately reveal the preferences of stakeholders. It is a simple but important point that is nicely illustrated in a context that is relevant to policymakers and vulnerable populations that are likely to be affected by these kinds of decisions.

Methodologically, I think that the authors would have been better served with a conjoint analysis, which is designed to more precisely estimate these kinds of tradeoffs. I would ask them to conceptually replicate the initial study with that method to ensure that the findings hold under a more typical analysis. This would also offer more precise estimates about the tradeoffs that would be pragmatically useful for policymakers (if they did use numbers from the papers to make decisions) and would also confirm that the results are not an artifact of the particular design being used.

(Remarks on code availability)

Version 1:

Reviewer comments:

Reviewer #1

(Remarks to the Author)

I read the revised manuscript with great interest. The authors have addressed my concerns and comments, and I am happy to recommend publication for this paper. I hope this paper will be read widely and considered in welfare discussions and public policy circles.

(Remarks on code availability)

Reviewer #2

(Remarks to the Author)

The new version is clearly an improvement and we appreciate the effort that has been made to address the criticisms and even carry out an additional study, which did address our main methodological concern regarding the work.

While we see many improvements, we still would like to push the authors a bit more. Our chief remaining critique regards the strength and certainty of some of the scientific claims made in the paper. We want to emphasize that it has nothing to do with whether we think the work is relevant or valuable - which we do -, nor is this meant as a suggestion to bring in more information to support current framing. Instead, we suggest substantially more nuanced phrasing regarding ascribing relevance to the findings within the complexities of the public sector. We provide some examples below, but we encourage the authors to go through the whole manuscript.

1) The framing of a necessity for designing AI involving public and or stakeholder preferences, confounding the population needs, population desires, and appropriate governmental design considerations, is appealing, but not warranted based on the experimental studies. Examples include line 58-61, "As a...public preference," 334-336, "governments must...the government," Line 377-380, "second, we...claimant population." 29-30 "neglect the needs of vulnerable groups" line 350-352" however, using... Population effectively". Though we may or may not agree with these notions, claiming scientific necessity for stakeholder involvement as done presently seems to overstate what can be concluded based on the data. Perhaps you could state that if policymakers were interested in designing to meet claimant desires, they may not rely solely on popular opinion and may have trouble empathizing altogether.

2) The notion of public preference – including the speed accuracy trade-offs – is directly used as a design requirement rather than an evaluative measurement in ways currently suggested by the article. Examples include line 48-58 (multiple examples) "for example... to award." line 75-78 "concerns about... AV behavior". Though we do not deny the value of understanding speed/accuracy trade-offs or say that public preference is fully disjunct from design considerations, we do not see how the provided examples are directly reflective of public preference. Additionally, the examples on speed/accuracy are seemingly less indicative of public opinion on speed/accuracy used in design and seem more so post hoc justification of the exemplified government programs.

We do not suggest adding additional text to address these concerns. Instead, we suggest altering the tone and hedging of existing claims to be significantly more nuanced. This may then be used to more appropriately highlight what we see as the strengths of the study: 1) the heterogeneous (in)ability of stakeholders to emphasize 2) the ability all stakeholders show in reasoning about trade-offs 3) suggesting how such data could be valuable if and only if there is a need or desire to have the voices of these vulnerable stakeholders heard.

3) Regarding the new study, we are impressed with the hard work put in to improve this article. As a side note, the argument mentioned in rebuttal that you are simply following the reporting style regarding speed/accuracy of government for preceding studies makes sense. It could be used in the article if it is not already. A clarifying improvement to presenting numerical findings would be to display for all four groups (Claimants perspective claimants, Claimants perspective non-claimants, ...

etc.) the acceptable trade-off in the overall terms you phrased them (0.2 lower accuracy for 1% speed increase).

(Remarks on code availability)

Reviewer #3

(Remarks to the Author)

(Remarks on code availability)

Reviewer #4

(Remarks to the Author)

I appreciate the authors' responsiveness to my suggestion for a new study. Thank you. The conjoint results support the authors' predictions with additional compelling evidence and are helpful in gauging the relative effects of speed and accuracy on AI acceptance.

I have a few additional minor points about places where the manuscript could be strengthened to clarify the results.

1. In Figure 6, it would be helpful in the left panel (6A) to illustrate the point estimates of the conjoint results for claimants and non-claimants. These comparisons seem as important, if not more important, than the illustration of the bias in perspective-taking across groups.

2. In Experiments 1, 2, and 3, there are no direct statistical comparisons between the degree of bias in the perspective-taking measure across claimants and non-claimants. Taking that into consideration, I think the following claim in the discussion isn't clearly communicating the results of the findings and could be misconstrued,

"Data revealed a further complication: asymmetric insights between claimants and 353 non-claimants. Claimants could provide relatively unbiased estimates of the 354 preferences of non-claimants, but non-claimants failed to do the same, even in the 355 presence of financial incentives."

I request that the authors either provide such analyses in each study or temper this claim.

3. In experiment 2, knowing each correlation between attributes and loss of trust in government for both the claimant and non-claimant groups would be helpful. A Williams test could compare the correlations with each attribute, revealing if speed or accuracy has a larger effect. My guess is that accuracy tradeoffs would result in a larger drop in trust than speed.

All in all, this is a nice paper that makes an important and timely point.

(Remarks on code availability)

Version 2:

Reviewer comments:

Reviewer #2

(Remarks to the Author)

We thank the authors for making "the extra mile" by addressing our final comments. We think that the paper is now ready for publication. The paper will be an excellent contribution to the debate on algorithmic decision-making in the public sector.

(Remarks on code availability)

Reviewer #3

(Remarks to the Author)

I co-reviewed this manuscript with one of the reviewers who provided the listed reports. This is part of the Nature

Communications initiative to facilitate training in peer review and to provide appropriate recognition for Early Career Researchers who co-review manuscripts.

(Remarks on code availability)

Reviewer #4

(Remarks to the Author)

The authors have addressed my final concerns and I believe the manuscript reports important and timely results.

(Remarks on code availability)

[Reviewer 1]

[Comment 1] This is a very interesting and timely paper on the perceptions of the use of AI in social welfare systems. The authors use two large samples (one from the US and the other from the UK, the latter pre-registered) to study the acceptance of trade-offs between speed and accuracy in the use of AI in the welfare processing system. The whole sample is, on average, willing to trade off speed at a loss of accuracy. However, subgroups including claimants and vulnerable populations are less inclined to exhibit such a trade-off and generally oppose the use of AI, especially as accuracy declines. This may seem unsurprising in hindsight, but one could have also imagined that the improved efficiency of AI (even at a slight cost of mistakes) could appeal to an audience in need for such benefits quickly.

This paper is well written, the methods are solid, the analysis is robust, and the results are both of general interest and nuanced in important ways. I like the figures in particular, they clearly articulate the findings, also to a broader audience.

This paper will provide impetus for other scholars to work on these important questions in the future (including with understudied populations such as claimants), but it also has immediate implications for policymakers and the welfare sector – this is critical since we are at a time when we need timely and robust research to inform policy-making around AI.

We thank the reviewer for the positive feedback and additional helpful comments. We are especially glad that the reviewer sees and appreciates the relevance to a broad audience.

[Comment 2] I have the following comments and suggestions for the authors:

The asymmetry between claimants' and non-claimants' perspective taking (or lack thereof) is interesting. Claimants, due to their vulnerable status, are an understudied population and I'm glad that you made the effort to look into this population's perspective (as, obviously, they would be most affected by the use of AI in this domain).

When you introduce the UK study (end of page 2), it may be worth pointing out as another "difference" to the US study that you recruited a targeted sample of claimants and non-claimants. Because it is so intentional, I believe it is worth mentioning as a deliberate methodological change to your design in this study and the sample size to reach meaningful conclusions.

At the beginning of the section “Study 2 Results”, we now highlight this information (p.8):

To replicate the results obtained from the US representative sample, this study collected data from $N = 1,462$ participants in the UK. Unlike the US representative-sample study, which had 20% claimants, we recruited an equivalent number of claimants and non-claimants in the UK. This sample size with a balanced composition of claimants and non-claimants can help consolidate our pre-registered hypothesis on the asymmetry in perspective-taking.

[Comment 3] In general, I would appreciate having the sample sizes for each panel (and each subgroup, e.g. claimants and non-claimants, where appropriate), either in the figure legends or inside the graphs.

This information is added in corresponding figure legends.

[Comment 4] A minor observation from your results in Study 2 might be worth drawing out: when AI has no accuracy loss (i.e. 0 on the x-axis) but there are also no speed gains (0 weeks faster), the whole sample seems to be on average "indifferent" between human and AI (and, line with earlier results, group 3 still prefers human across the board). This may seem unimportant at the surface but from a public policy position (and taxpayer perspective!), it would be so much more cost-effective to use AI over human – in this case without the real benefits or drawbacks of AI, except that, once running, the AI saves a lot of labour costs. Of course, we are a long way away from a "perfect error-free AI", but policy-makers and scholars alike might be interested in seeing that there is less of an "algorithm aversion" than has previously been documented.

As suggested, we report the preference when AI has no accuracy loss and speed gain, as compared to the mid-point of the preference scale (meaning “indifferent” between humans and AI). However, the difference was significant. As we add in the “Study 2 Results” (p.8):

Notably, when welfare AI demonstrates comparable performance (i.e., 0 week faster and 0% more error), people were still in favor of humans making welfare decisions ($M = 45.4$, $SD = 28.7$; $t = 4.36$, $p < .001$).

[Comment 5] In the discussion section, I think it would be good to make explicit a counterfactual not currently discussed: assume you don't use AI and just rely on humans – they, too, make errors. So, what are the welfare gains from using AI that

avoid the opposite risk, i.e. that human decision-makers in the absence of AI would make an inaccurate decision – but without benefit of speed! Put differently, it is worth remembering for scholars and policy-makers alike that neither humans nor AI are infallible, so future research may want to take a closer look at what people would prefer if they could only choose between fallible AI and human options.

We appreciate this point for discussion. In the revised Discussion section, we now add that (p.15):

[T]he current research focused on one critical type of tradeoff for the deployment of AI in welfare systems: the speed-accuracy tradeoff. We operationalized AI inaccuracy as additional false rejections compared to human conditions, and found that claimants were much more averse than non-claimants to welfare AI programs and their mistakes. However, these findings do not imply that AI programs should not be launched before they become perfect. They also do not suggest that human decisions are error-free, or that AI always makes more mistakes than humans in welfare decisions. In reality, AI- and human-dominant government systems may face different challenges. For example, AI can be hyper-vigilant about anomalies^{8,9} and seen as inflexible in self-corrections^{13,14}. In contrast, human public servants may discriminate against particular social groups, and such biased judgments may vary from person to person and induce inconsistencies and unfairness in welfare payments^{44,45}. Therefore, future research may explore public opinions for other tradeoffs, such as different types of inaccuracy introduced by AI versus human welfare systems.

[Comment 6] Personally, I'm less convinced of the value of the pre-registration given that you already have two sizeable samples from two different countries and, importantly, find largely similar results: given that most of your estimates with the same analyses are pretty consistent across the two studies (except the difference of the accuracy of perceptions of non-claimants between the two studies), I think the pre-registration isn't really telling us much that we didn't already expect to see from pooled $n=2,500$ and a conceptual replication. However, I appreciate the transparency in including the directional hypotheses you made for study 2 as well as the fact that some of those hypotheses did not match the empirical findings in study 2. That said, I think not having these predictions in the main text would improve readability (at no loss of accuracy!). (And these pre-registered hypotheses could just as well be mentioned in the Materials and Methods section.) But this is a matter of preference.

To increase readability, we removed the descriptions of our pre-registered hypotheses from the section “Study 2 Results”. But to maintain transparency of our pre-registered analyses, we mentioned them later in the Methods section (p.19):

We also pre-registered three predictions. First, we expected that accuracy losses would matter more to participants than speed gains. Practically speaking, we expected that participants would value one experimental unit of speed gains (1 week) less than one experimental unit of accuracy loss (5 percentage points). This prediction was not supported since participants tolerated a 5 percentage point accuracy loss for a 1-week speed gain. Second, we expected to identify subgroups of participants with different patterns of trade-offs between speed gains and accuracy losses. This prediction lacked strong support since we did not find a particular number of profiles that significantly outperformed others (see the Supplementary Information). Third, we expected that non-claimants would not be good at predicting the preferences of claimants. The third prediction was confirmed.

[Reviewer 2]

[Comment 1]

Key results

The study's key result is showcasing the heterogeneity between claimants of welfare and non-claimants of welfare in how acceptable they find decreases in accuracy for a faster welfare service when replacing decision-making with AI. Welfare claimants were far less willing to trade off speed for accuracy. Additionally, non-welfare claimants were found less capable of taking claimants' perspectives. The authors combined this with the fact that non-claimants make up the majority to conclude that by averaging opinions in a majority rule system, we – as a society – may falsely believe a certain AI implementation is desirable, despite chief stakeholders potentially disagreeing.

General feedback

Overall, the study is technically well done and provides some interesting results. The findings are documented well and pleasantly written up. As far as we are aware the contribution is novel. However interesting, the overall significance of the results for a general audience – as could be expected for this journal – seems somewhat narrow. In addition, there may also be some issues with the design of the experiment. We will explain these points below, in addition to some more specific points of feedback.

Line 52: We find the phrasing of “this balance must be informed by public preference” somewhat questionable. The assumption that majority preference results in changes in

AI design biased against vulnerable groups seems overly simplistic and not grounded in current theory. While it may be true that vulnerable stakeholders may be missed out on in AI design, there are more complicated mechanisms at work that are overlooked in this paper. First, if majority preferences have an effect this is very indirectly: e.g., this may be caused by political decision-making that limits the autonomy of welfare case workers to grant welfare, but this often requires legal changes which only indirectly leads to AI design changes. Second, when developing AI systems, managers and developers tend to have a technical-rational perspective and thereby overemphasizing accuracy and efficiency while missing out on taking stakeholder perspectives at large (For this and our other points on value-sensitivity and how this is translated into practice, see for example Fest, I., Schäfer, M., van Dijck, J., & Meijer, A. (2023). Understanding Data Professionals in the Police: A Qualitative Study of System-Level Bureaucrats. Public Management Review, 0(0), 1–21. <https://doi.org/10.1080/14719037.2023.2222734>).

We thank the reviewer for the helpful comments. In response to this first point, as suggested, we have added a paragraph in the Introduction to elaborate on the various stakeholders and complex mechanisms involved in the design of government AI systems, incorporating the Value-Sensitive Design (VSD) framework to highlight the integration of diverse perspectives in AI development (p.3):

Designing AI systems for social good requires not only technological progress but also the integration of a broader set of ethical, legal, and societal considerations, which necessitates incorporating the perspectives of various direct and indirect stakeholders¹³. Caseworkers, developers, and program managers can develop an understanding of the needs and pain points of users of government AI systems through exposure to diverse user cases and civic discussion^{14,15}. However, their technical-rational perspective may lead them to overemphasize certain performance metrics while overlooking the perspectives of the general public^{14,16}. Public deliberation also plays a crucial role in ensuring that AI systems align with societal values and are perceived as fair and legitimate. While public preferences may not directly dictate AI design choices, they influence the legal and regulatory environment in which AI systems operate, shaping AI design and deployment through political decision-making processes.

[Comment 2] Line 210-211: "In summary...of stakeholders". We think this summation states a stronger conclusion than can be inferred from the article. The article convincingly shows how following popular opinion can result in the main stakeholders of a deployed AI system not seeing their design desires reflected, so perhaps such a

system fails in giving voice to those it aims to help. However, even if design specifications are inspired by popular opinion, the study does not show that this results in a degradation of government trust at large. It is easy to imagine that the efficiency gains implied by the article result in e.g. a cheaper government and more labour available elsewhere, which may be a sufficient reason for the majority to still prefer to push through their opinion and improve their trust.

We rephrased the summative sentence and included the nice illustration in the reviewer's comment (p.14):

In summary, average responses may not capture the divergent preferences of stakeholders in welfare AI systems. It is easy to imagine that efficiency gains – such as a more cost-effective government and increased labor availability – could be enough to convince the majority, non-claimant population to accept welfare AI systems and improve their trust. However, using this data to calibrate AI in welfare systems could overlook the voices and needs of those directly affected, and ultimately fail to serve this targeted population effectively.

We further discuss the potential mechanisms underlying public trust, acknowledging that they are not fully addressed in the present research (p.15):

[W]hile we identified strong correlations between tradeoff preferences and trust in government, we did not directly test how public trust evolves with full versus partial transparency. The real-life mechanisms underlying public trust are more complicated than this study addressed^{40,41}. For example, beyond information from governmental agencies, public opinion is increasingly influenced by exposure to the suffering of vulnerable individuals due to algorithmic mistakes on social media and news platforms^{7,10}. Future research should systematically examine how trust in government changes when different aspects (e.g., technical, anecdotal, societal) of welfare AI systems are communicated.

[Comment 3] Line 221-227: This paragraph seems contradictory. Non-claimants are unable to take the perspective of claimants even when they try, so the solution is to actively engage with claimants. However, presumably, those engaging with them are predominantly non-claimants, so how does this solve anything? I would suggest putting these vulnerable stakeholders in a position of co-design or letting subsequent research find ways to allow their perspectives to be taken.

We clarified the point of “the need to actively engage with claimants” and now state that (p.14):

Our results thus underline the need to involve potential claimants in the co-design process, or develop technical solutions that incorporate their perspectives and preferences when configuring AI in welfare systems – rather than to assume that their preferences are well-understood or can be understood through empathetic perspective-taking.

[Comment 4] Line 228: The argument for transparent communication is still underdeveloped. Several questions remain unanswered, e.g. Does this solve the issue addressed by this study, and how? What is the target audience of the communication and what would the communication entail?

We expanded this paragraph in Discussion and responded to these questions as below (p.14-15):

Our results also shed light on the potential for transparent communication about the design choices of welfare AI systems, especially in the political decision-making processes involving the general public. First, we demonstrate that people can systematically evaluate the benefits and costs of deploying welfare AI systems, rather than focusing solely on negative features. Public disclosure of AI inaccuracies does not simply lead to criticism and pushback; people also value the accompanying speed gains and relative accuracy improvements, possibly over time. Second, we offer scientific support for public communication strategies when welfare AI designs must prioritize the preferences of a small subgroup of claimants over the majority, non-claimant population. These design choices can be clearly justified by the realities of heterogeneous preferences and asymmetrical insights.

[Comment 5] Data, methodology & validity

Overall, the methodological approach is well documented. The data is freely available and made very accessible through the provided description. For the most part, applied methodology is sound and should provide some level of validity to real-world contexts. Yet we also have the following more specific points:

We have already brought up our concerns over the external validity of this study in the general section. Specifically, we question both whether the popular opinion is translated into design specifications of AI or other political and administrative decisions, as well as whether a speed/accuracy tradeoff is currently (widely) used as a pre-specified design consideration for public sector AI use.

In response to the first point (*“whether the popular opinion is translated into design specifications of AI or other political and administrative decisions”*), besides the new introduction of the various stakeholders and complex mechanisms involved in the design of government AI systems (see reply to Comment 1), we use the example of autonomous vehicles to illustrate how AI designs are informed by public preferences (p.3):

A prominent example here is the public engagement in the design and regulation of autonomous vehicles (AVs). Concerns about disproportionate harm to vulnerable road users and ethical decision-making in crash scenarios have gained significant public and media attention, prompting policies focused on transparency, explainability, and accountability of AV behavior ¹⁷.

Regarding the second point (*“whether a speed/accuracy tradeoff is currently (widely) used as a pre-specified design consideration for public sector AI use”*), we provide more contexts based on the government reports and third-party investigations that helped validate our research question and experimental design (p.2-3):

[S]peed and accuracy are important performance metrics in government services and their public communications about innovative welfare AI systems. For example, before the Royal Commission into the notorious Robodebt scheme in Australia, the government annual report 2019-2020 stated that “the agency automated the assessment and processing of most claims for services”, “we processed 1.3 million JobSeeker claims in 55 days, a claim volume normally processed in two and a half years”, and “the agency recorded 276,589 feedback contacts...dissatisfaction with a decision, outcome or payment, including waiting too long, not receiving a payment, and rejection of an application or claim (32.1 percent)” ¹¹. Similarly, the UK Department for Work and Pensions Annual Report stated that with “[i]ncreased use of data analytics and greater automation”, they had “146,000 claims checked by Enhanced Checking Service”, among which “87,000 check result in change to award” ¹².

As illustrated in these two examples, speed and accuracy are realistic metrics used in political and administrative communications about welfare AI programs.

[Comment 6] One important concern regards the vignette design, and in particular, how speed and accuracy are presented to participants. Speed is presented with a baseline and improvements are an absolute amount (e.g. it takes 4 weeks, AI makes it 3 weeks faster). However, the accuracy improvement is presented as a relative percentage without a baseline (AI helps detect fraud and leads to a 20% higher chance of rejection).

First off, this means that if claimants and non-claimants are simply divergent in the baseline they assume/experience in existing rejection, you are measuring that difference. For example, if claimants already experience a 50% false rejection rate, then any decrease in accuracy is very undesirable. However, if non-claimants assume a 1% false rejection rate, even a 100% higher false rejection rate is somewhat fine if it speeds up the process. We would argue that you would require some knowledge of what people across these groups think the baseline accuracy is to conclude. This can then still be a conclusion about divergence in participants taking one another's perspective, though the reason may be rooted in a lack of knowledge instead of an inability to take perspective. Secondly, people in general are not great at dealing with estimates in the way you present them. Even if the baseline of let's say 10% rejection is assumed by everyone and we are given a 2-week speed boost, a 20% increase in false rejection seems like a very significant increase with 2 weeks not sounding like that much at all, but in absolute terms this would only result in the total rejection being 12% or a 2% difference in absolute terms while it would be a 50% increase in speed. Just by phrasing it in terms consistent with one another, we would argue the trade-off sounds much more reasonable without changing at all (For both these points, work by the behavioural economist Gigerenzer such as "The bias bias in behavioural economics" could provide some additional context).

We appreciate the reviewer's constructive feedback on the potential confounds in our operationalizations of speed gains and accuracy losses. As shown in our reply to Comment 5, we manipulated speed information by time and accuracy information by percentage to be consistent with real-life governmental reports about welfare AI programs.

That said, we also integrated these considerations into the design of a new experiment ("the US balanced-sample conjoint study" in the revised manuscript), to consolidate the results of the first US representative-sample study (see p.11-13 for Results and p.19-20 for Methods). Below is a summary of how the new experiment design addresses the comments:

- To address the lack of baseline descriptions for accuracy information, the new experiment presents baselines for both speed and accuracy ("Typically, a public servant processes such applications within 40 working days and has a false rejection rate of 30%").
- To address the potential difficulty of information interpretation, the new experiment presents speed and accuracy information in both absolute and comparative terms. For example, in the condition where "compared to public servants, the AI program usually makes a welfare decision with a 10% shorter

waiting time and a 10% higher chance of falsely rejecting [your/the person's] application", it is also noted that "10% shorter waiting time = 36 working days; 10% higher chance of false rejection = 33% false rejections".

As such, the new experiment perfectly aligns the manipulation of speed gains and accuracy losses, by presenting human baselines, AI performance in absolute and comparable terms, and matched AI performance intervals (0/10/20/30/40/50% shorter waiting time and 0/10/20/30/40/50% higher chance of false rejection) for speed and accuracy information. Integrating the suggestions of Reviewer 4, the new experiment is implemented in a choice-based conjoint design. That is, instead of Likert-scale responses to preferences for AI or humans, participants make binary choices between pairs of AI with different speed and accuracy combinations. With this new study, we conceptually replicated previous findings on heterogeneous preferences and asymmetrical insights between claimants and non-claimants.

[Comment 7] Line 272: It was a bit unclear as to why revision of answers in the survey was not allowed anymore. Perhaps provide a reason for this.

We realize that our previous description was not entirely clear in that respect. Participants could always revise their answers on the current page (by moving the slider around) before validating their answers (by clicking to proceed to the next page). In the main survey, we simply disabled the possibility of going back to previous pages or revising validated answers to previous tradeoff scenarios. As we now explain (p.17):

These exercise trials aim to familiarize participants with the experiment paradigm. Therefore, we gave participants a chance to review the example stimuli and their corresponding answers, and calibrate their answers before moving to the 36 official test rounds. The official test rounds no longer allowed revisions, including going back to previous pages or revising validated answers to previous tradeoff scenarios, which was clearly explained to participants at the end of the training session.

[Comment 8] Line 310-314: There are methodological discussions about how well financial incentives work. Perhaps those less well-off (the claimants) are more motivated to answer correctly (or even seek out information to allow them to answer correctly).

It is reasonable to assume that claimants are likely more motivated than non-claimants to answer correctly when given the same amount of incentives. However, determining the precise level of additional incentives that would equalize motivation between the two

groups is inherently difficult, as excessive rewards could unintentionally distort engagement. In the experiment context, we deem the basic payment for study participation as an objective benchmark for setting incentives for accurate perspective-taking. Since all participants received equal compensation for completing the study, their baseline motivation to earn additional rewards should be roughly comparable.

Moreover, our findings are not solely dependent on the incentive structure. Both the first US study, which used a representative sample, and the new US study, which had a balanced composition of claimants and non-claimants, were conducted without additional incentives for accuracy. Across these three studies, we observe consistent evidence of asymmetrical perspective-taking, irrespective of whether accuracy was incentivized. This convergence strengthens our conclusion that the effect is not merely an artifact of financial motivation but reflects a deeper asymmetry in perspective-taking.

[Comment 9] Mostly of the following points reiterate our comments above, but the following are some of our chief suggestions for improvement and extension:

- *A more nuanced argumentation of how the preferences of stakeholders end up inspiring AI design, as one may, for example, find in work on value-sensitive design.*

This point is addressed in our reply to Comment 1.

- *A more intricate design. For example, perhaps having the option to appeal to a human decision-maker might make it very worthwhile to accept a very fast but inaccurate AI system, which could be combined with how discernible it is that you are falsely and not rightly rejected.*
- *An extension regarding perspective taking which performs a comparison of speed and accuracy in absolute terms with the presence of a baseline*

As detailed in our reply to Comment 6, we have implemented a more intricate design in the new study, with human baselines for both speed and accuracy information and AI performance in comparative and absolute terms. However, we did not incorporate the scenario of appealing to human decision-makers since it has been tested in the UK balanced-sample study where “[participants] were then randomly assigned to a no redress or a human redress condition, which differed on whether claimants could appeal to public servants” (p.18). We found that (p.8):

Even though participants in the human redress condition believed in the chance to appeal in our manipulation check ($\beta = 0.37$, $p < .001$; vs. the redress

condition), this clarification did not impact trade-off preferences ($\beta = 0.03$, $p = .210$).

- *An extension that shows the discussed issue is based on lack of knowledge (claimants know the inaccuracy is higher than the non-claimants think it is), perspective taking (non-claimants are unable/unwilling to empathize with the position of claimants though they share knowledge) or perhaps both*

Here, we discuss the lack of knowledge about human and AI accuracy separately. Regarding human accuracy, it is indeed possible that non-claimants underestimate the inaccuracy of human-based welfare decisions. However, this is not directly related to our main research question, which focuses on preferences for AI accuracy. To account for this potential confound, we ensure that both claimants and non-claimants were exposed to a consistent human accuracy baseline, as detailed in our response to Comment 6.

In contrast, AI accuracy is an integral component of our experimental manipulation. Before measuring their preferences or choices, we provide claimants and non-claimants with the same standardized information about AI accuracy. As a result, any observed biases in non-claimants' perspective-taking cannot be attributed to differences in their state of information about AI accuracy.

- *Why only consider the speed/accuracy tradeoff in the chosen direction? Perhaps claimants would first want both faster and more accurate systems before it is worthwhile, suggesting support for AI introduction in this group is dependent on other factors. Similarly, perhaps claimants are happy to take a slow-down for higher accuracy. This could be a worthwhile tradeoff for example if normally you are rejected so often that a slightly slower system is worth it. The preceding may not be a realistic scenario for AI introduction, but could steer policymakers to prioritize accuracy at the cost of speed.*

Regarding the first point (“perhaps claimants would first want both faster and more accurate systems before it is worthwhile, suggesting support for AI introduction in this group is dependent on other factors”), we agree that claimants want both faster and more accurate systems. This is part of the findings in our work: people appreciate both speed and accuracy improvement, but in reality, there are often trade-offs between these two factors. As for other factors, we examined human redress in the UK balanced-sample study (i.e., the chance to appeal to public servants) and did not find a significant effect. We now also acknowledge this possibility following the relevant discussion of the new US conjoint study (p.13):

[W]hile claimants accurately predict the importance of speed for non-claimants, non-claimants overestimate the importance of both speed and accuracy for claimants. These findings conceptually replicate previous studies, and suggest that claimants may prioritize other factors that non-claimants fail to recognize.

Regarding the second point (*“perhaps claimants are happy to take a slow-down for higher accuracy. This could be a worthwhile tradeoff for example if normally you are rejected so often that a slightly slower system is worth it”*), we agree that claimants may be willing to trade off in this direction for welfare decisions in general. However, as detailed in our reply to Comment 5, this does not seem to be a realistic scenario for welfare AI systems, which are unlikely to be slower than humans.

References:

For the most part, the provided references are appropriate. Beyond what was mentioned in the general section, we have the following points:

- *Line 45 & line 199: As far as we can tell, the references to sources 4 and 5 seems out of place. Neither article referenced is about the preceding statement concerning the main promise of introducing AI in decision-making for welfare supposedly speeding up decision-making. At best, both articles make some mention of the efficiency and effectiveness of AI in government at large or refugee placement specifically. Yet for a generic point on efficiency & effectiveness concerning AI in the public sector, other sources would probably be more appropriate. Neither of the provided sources specifically covers speeding up decision-making through AI or this being considered the main promise for its introduction in welfare.*

We agree that these two references spoke broadly to the welfare effects of AI in government. We replaced them with references that directly analyzed AI programs used to distribute social welfare benefits.

[Reviewer 3]

We thank the reviewer for the co-reviewing and helpful comments.

[Reviewer 4]

The authors explore preferences for speed-accuracy tradeoffs between people who do and do not claim social benefits, with the aggregate sample in the US study reflecting the census data (i.e., nationally representative).

The key finding of the paper is that non-claimants overestimate the degree to which claimants prioritize convenience over accuracy. This illustrates how nationally representative samples may fail to accurately reveal the preferences of stakeholders. It is a simple but important point that is nicely illustrated in a context that is relevant to policymakers and vulnerable populations that are likely to be affected by these kinds of decisions.

Methodologically, I think that the authors would have been better served with a conjoint analysis, which is designed to more precisely estimate these kinds of tradeoffs. I would ask them to conceptually replicate the initial study with that method to ensure that the findings hold under a more typical analysis. This would also offer more precise estimates about the tradeoffs that would be pragmatically useful for policymakers (if they did use numbers from the papers to make decisions) and would also confirm that the results are not an artifact of the particular design being used.

We thank the reviewer for the positive feedback and further insightful comments. As suggested, we conducted a new conjoint study in the US with a balanced sample of claimants versus non-claimants (N = 800). Integrating Reviewer 2's suggestion, we manipulated both speed gains and accuracy losses by percentage and provided baseline information for human accuracy. Specifically (p.11):

In the previous two studies, participants indicated preferences for individual AI programs, featuring speed gain by week and accuracy loss by percentage. This study aims to conceptually replicate previous findings in a choice-based conjoint experiment, where participants (1) select one of two AI programs presented in pairs and (2) evaluate the information of both speed gain and accuracy loss by percentage. We recruited a balanced sample of claimants and non-claimants from the US (N = 800). Each participant made binary choices for 30 pairs of AI programs, varying on speed gain (0%/10%/20%/30%/40%/50% shorter waiting time, as compared to a baseline of 40 working days if handled by public servants) and accuracy loss (0%/10%/20%/30%/40%/50% higher chance of false rejection, as compared to a baseline of 30% false rejection rate if handled by public servants).

In general, our main findings on heterogeneous preferences and asymmetrical insights are conceptually replicated by adopting the conjoint design and analysis (see p.11-13 for Results and p.19-20 for Methods in the revised manuscript). To summarize:

[O]ur conjoint study reveals that US citizens, comprising a balanced sample of claimants and non-claimants, are willing to trade off a 0.2 percentage point reduction in AI accuracy for each 1 percentage point increase in speed. Beneath this overall pattern, however, non-claimants respond more positively than claimants to speed gains of welfare AI programs. We further show the asymmetry in perspective-taking: while claimants accurately predict the importance of speed for non-claimants, non-claimants overestimate the importance of both speed and accuracy for claimants. These findings conceptually replicate previous studies, and suggest that claimants may prioritize other factors that non-claimants fail to recognize. By employing a different design and analysis strategy, the conjoint study further strengthens the robustness of previous results, demonstrating that they are not tied to a specific experimental setup.

[Reviewer 1]

I read the revised manuscript with great interest. The authors have addressed my concerns and comments, and I am happy to recommend publication for this paper. I hope this paper will be read widely and considered in welfare discussions and public policy circles.

We thank the reviewer for taking the time to review our revised manuscript and for all the helpful feedback.

[Reviewer 2]

The new version is clearly an improvement and we appreciate the effort that has been made to address the criticisms and even carry out an additional study, which did address our main methodological concern regarding the work.

While we see many improvements, we still would like to push the authors a bit more. Our chief remaining critique regards the strength and certainty of some of the scientific claims made in the paper. We want to emphasize that it has nothing to do with whether we think the work is relevant or valuable - which we do -, nor is this meant as a suggestion to bring in more information to support current framing. Instead, we suggest substantially more nuanced phrasing regarding ascribing relevance to the findings within the complexities of the public sector. We provide some examples below, but we encourage the authors to go through the whole manuscript.

*1) The framing of a necessity for designing AI involving public and or stakeholder preferences, confounding the population needs, population desires, and appropriate governmental design considerations, is appealing, but not warranted based on the experimental studies. Examples include line 58-61, "As a...public preference," 334-336," governments must...the government," Line 377-380, "second, we...claimant population."29-30 "neglect the needs of vulnerable groups" line 350-352" however, using.... Population effectively". Though we may or may not agree with these notions, claiming scientific necessity for stakeholder involvement as done presently seems to overstate what can be concluded based on the data. **Perhaps you could state that if policymakers were interested in designing to meet claimant desires, they may not rely solely on popular opinion and may have trouble empathizing altogether.***

We thank the reviewer for the thoughtful feedback and for recognizing the improvements in our manuscript. As suggested, we have refined the relevant statements to ensure they are more precise and aligned with the scope of our data. Below, we present the original and revised texts in the order they appear in the manuscript:

- a) Original: “This suggests that policy decisions influenced by the dominant voice of non-claimants may neglect the actual needs of vulnerable groups.” (p.2)
Revised: “This suggests that policy decisions aimed at supporting vulnerable groups may need to incorporate minority voices beyond popular opinion, as non-claimants may not easily understand claimants’ perspectives.”
- b) Original: “As a result, government agencies that seek to deploy welfare AI systems must strike a careful balance between speed gains and accuracy losses, and this balance must be informed by public preferences.” (p.3)
Revised: “Although these reports do not explicitly state whether and how welfare AI systems trade off between speed and accuracy, government agencies that seek to deploy welfare AI systems in an acceptable and trustworthy way may benefit from carefully considering public preferences when balancing speed gains and accuracy losses.”
- c) Original: “Governments must carefully balance these trade-offs to maintain public trust^{18,29}. Indeed, we found that the acceptability of this balance to participants was closely tied to their resulting trust in the government.” (p.14)
Revised: “Governments deploying welfare AI systems may need to navigate these trade-offs carefully, particularly given their potential impacts on public trust^{18,29}. Our findings also suggest that the acceptability of these trade-off decisions is strongly correlated with public trust in the government.”
- d) Original: “However, using this data to calibrate AI in welfare systems could overlook the voices and needs of those directly affected, and ultimately fail to serve this targeted population effectively.” (p.14-15)
Revised: “However, if governments aim to align welfare AI systems with claimants’ preferences, they may need to look beyond aggregate public opinion, as it does not necessarily capture the perspectives of those directly affected.”
- e) Original: “Second, we offer scientific support for public communication strategies when welfare AI designs must prioritize the preferences of a small subgroup of claimants over the majority, non-claimant population. These design choices can

be clearly justified by the realities of heterogeneous preferences and asymmetrical insights” (p.15)

Revised: “Second, we offer scientific support for public communication strategies when welfare AI systems prioritize the preferences of a small subgroup of claimants over the majority, non-claimant population. These decisions about whose values and preferences AI aligns with – often referred to as ‘the alignment problem’⁴¹⁻⁴³ – can be justified by the realities of heterogeneous preferences and asymmetrical insights in the context of welfare decisions.”

2) The notion of public preference – including the speed accuracy trade-offs – is directly used as a design requirement rather than an evaluative measurement in ways currently suggested by the article. Examples include line 48-58 (multiple examples) “for example... to award.” line 75-78 “concerns about... AV behavior”. Though we do not deny the value of understanding speed/accuracy trade-offs or say that public preference is fully disjunct from design considerations, we do not see how the provided examples are directly reflective of public preference. Additionally, the examples on speed/accuracy are seemingly less indicative of public opinion on speed/accuracy used in design and seem more so post hoc justification of the exemplified government programs.

We do not suggest adding additional text to address these concerns. Instead, we suggest altering the tone and hedging of existing claims to be significantly more nuanced. This may then be used to more appropriately highlight what we see as the strengths of the study: 1) the heterogenous (in)ability of stakeholders to emphasize 2) the ability all stakeholders show in reasoning about trade-offs 3) suggesting how such data could be valuable if and only if there is a need or desire to have the voices of these vulnerable stakeholders heard.

Indeed, the examples from government reports are included to illustrate how welfare AI programs are communicated to the general public and provide factual context for studying public preferences regarding speed-accuracy trade-offs. These examples are not intended to suggest that speed-accuracy trade-offs or public preferences directly dictate AI design considerations, which remain largely opaque in public communications. Likewise, the example of autonomous vehicles is used to illustrate how public opinion can shape the broader AI development trajectories through political decision-making processes – rather than as an immediate design specification. To clarify this intent, we have adjusted the surrounding text as follows (p.2):

Given the practical relevance of speed and accuracy for welfare decisions, these performance metrics are often highlighted in public-facing government reports after deploying AI systems, based on the implicit assumption that the general public values speed and accuracy in government services. ...Although these reports do not explicitly state whether and how welfare AI systems trade off between speed and accuracy, government agencies that seek to deploy welfare AI systems in an acceptable and trustworthy way may benefit from carefully considering public preferences when balancing speed gains and accuracy losses.

Additionally, we also understand that our use of the term 'design' may have been ambiguous, encompassing both technical configurations of AI systems ("public preferences may not directly dictate AI *design* choices") and broader ethical, legal, and societal considerations ("shaping AI *design* and deployment through political decision-making processes"). To avoid misunderstanding, we have replaced 'design' with more precise terms that better reflect how public preferences intersect with the broader AI development trajectories. Specifically, we made the following adjustments:

- a) "This work highlights the importance of stakeholder engagement and transparent communication in government deployment of AI, particularly in power-imbalanced contexts." (Abstract)
- b) "Developing AI systems for social good requires not only technological progress but also the integration of a broader set of ethical, legal, and societal considerations, ..." (p.3)
- c) "[P]ublic preferences ... shaping AI development and deployment through political decision-making processes." (p.3)
- d) "A prominent example here is the public engagement in the formulation and regulation of autonomous vehicles (AVs)." (p.3)
- e) "However, less is known about the extent of divergence in AI performance preferences and reconciliation between different perspectives and interests." (p.4)
- f) "Our results also shed light on the potential for transparent communication about the performance and alignment choices of welfare AI systems, especially in the political decision-making processes involving the general public." (p.15)

3) Regarding the new study, we are impressed with the hard work put in to improve this article. As a side note, the argument mentioned in rebuttal that you are simply following the reporting style regarding speed/accuracy of government for preceding studies makes sense. It could be used in the article if it is not already. A clarifying improvement to presenting numerical findings would be to display for all four groups (Claimants perspective claimants, Claimants perspective non-claimants, ...etc.) the acceptable trade-off in the overall terms you phrased them (0.2 lower accuracy for 1% speed increase).

We appreciate the reviewer's positive feedback and thoughtful suggestions. As suggested, we now explicitly state in the manuscript that "[t]he style of presenting speed and accuracy information closely follows relevant public-facing government reports", as noted in the legend of Fig. 1 An example of experimental stimuli (p.5) and the Method section (p.17).

We also provide the requested trade-off information in different experimental conditions. Specifically:

Overall, for each 1 percentage point increase in speed, claimants and non-claimants were willing to tolerate a 0.2 and 0.3 percentage point of AI accuracy loss, respectively. (p.11-12)

Overall, when taking the other group's perspective, claimants and non-claimants were willing to tolerate a 0.3 and 0.4 percentage point of AI accuracy loss, respectively, for each 1 percentage point increase in speed. (p.13)

[Reviewer 3]

We thank the reviewer for the co-reviewing and additional comments.

[Reviewer 4]

The authors explore preferences for speed-accuracy tradeoffs between people who do and do not claim social benefits, with the aggregate sample in the US study reflecting the census data (i.e., nationally representative).

I appreciate the authors' responsiveness to my suggestion for a new study. Thank you. The conjoint results support the authors' predictions with additional compelling evidence and are helpful in gauging the relative effects of speed and accuracy on AI acceptance.

I have a few additional minor points about places where the manuscript could be strengthened to clarify the results.

1. In Figure 6, it would be helpful in the left panel (6A) to illustrate the point estimates of the conjoint results for claimants and non-claimants. These comparisons seem as important, if not more important, than the illustration of the bias in perspective-taking across groups.

We thank the reviewer for the positive feedback on our new conjoint study and the additional comments. As suggested, we have provided the point estimates for claimants and non-claimants in the new Table 1 (p.13; also shown below). Figure 6 remains consistent with the style of previous figures.

Table 1. Comparisons of average marginal component effects (AMCEs) between claimants and non-claimants at each level of accuracy loss and speed gain.

	Accuracy Loss			Speed Gain		
	Claimants [95% CI]	Non-claimants [95% CI]	z (p)	Claimants [95% CI]	Non-claimants [95% CI]	z (p)
10%	-0.12 [-0.15, -0.10]	-0.11 [-0.14, -0.09]	-0.55 (.59)	0.01 [-0.01, 0.04]	0.06 [0.03, 0.09]	-2.36* (.02*)
20%	-0.25 [-0.28, -0.22]	-0.26 [-0.29, -0.24]	0.76 (.45)	0.05 [0.02, 0.08]	0.09 [0.06, 0.12]	-1.97* (.05*)
30%	-0.35 [-0.37, -0.32]	-0.39 [-0.41, -0.36]	2.04** (.04)	0.07 [0.04, 0.10]	0.12 [0.10, 0.15]	-2.74** (.01)
40%	-0.48 [-0.50, -0.45]	-0.53 [-0.55, -0.50]	2.70** (.01)	0.07 [0.04, 0.10]	0.16 [0.14, 0.19]	-4.71*** ($< .001$)
50%	-0.58 [-0.61, -0.56]	-0.64 [-0.66, -0.62]	3.30*** ($< .001$)	0.11 [0.08, 0.14]	0.19 [0.16, 0.22]	-4.09*** ($< .001$)

Note. $p < .05^$. $p < .01^{**}$. $p < .001^{***}$.*

2. In Experiments 1, 2, and 3, there are no direct statistical comparisons between the degree of bias in the perspective-taking measure across claimants and non-claimants. Taking that into consideration, I think the following claim in the discussion isn't clearly communicating the results of the findings and could be misconstrued,

"Data revealed a further complication: asymmetric insights between claimants and 353 non-claimants. Claimants could provide relatively unbiased estimates of the 354 preferences of non-claimants, but non-claimants failed to do the same, even in the 355 presence of financial incentives."

I request that the authors either provide such analyses in each study or temper this claim.

We agree with the reviewer that the original claim may not clearly communicate our empirical results. Instead of directly comparing the magnitude of bias between the two groups, our analysis strategy assessed whether a bias existed by comparing each group's estimates against the targeted group's actual preferences or choices. Since this approach was pre-registered in the UK study, we decided to maintain the original

analysis strategy rather than conducting new statistical tests. However, we have revised the claim to better reflect the available evidence (p.15):

“Data revealed a further complication: asymmetric insights between claimants and non-claimants. While neither group was perfectly accurate in understanding the perspective of the other, non-claimants were more likely to provide biased estimates of claimant’s preferences or choices, even in the presence of financial incentives.”

3. In experiment 2, knowing each correlation between attributes and loss of trust in government for both the claimant and non-claimant groups would be helpful. A Williams test could compare the correlations with each attribute, revealing if speed or accuracy has a larger effect. My guess is that accuracy tradeoffs would result in a larger drop in trust than speed.

All in all, this is a nice paper that makes an important and timely point.

As suggested, we have conducted Williams’s t-test for dependent correlations and incorporated the results in the revised manuscript (p.9):

Moreover, for both groups, accuracy losses had a significantly stronger correlation with the loss of trust in the government ($r = .41$ for claimants, and $r = .42$ for non-claimants), compared to speed gains ($r = -.21$ for claimants, $t = 40.8$, $p < .001$; $r = -.25$ for non-claimants, $t = 47.9$, $p < .001$).